

# Improving the Gravity Anomaly Map of French-Belgian Hainaut using Multi-Scale Fusion of New Gravity Acquisitions and Legacy Data in an Adaptive and Open-Source Gravimetric Processing Workflow.

Quentin CAMPEOL[1], Nicolas DUPONT[1] and Olivier KAUFMANN[1]

[1]Department of Geology and Applied Geology, University of Mons, Faculty of Engineering, 9, rue de Houdain, 7000 Mons, Belgium

*Correspondence to*: Quentin CAMPEOL (quentin.campeol@umons.ac.be)

**Keywords: Gravity Anomaly; Acquisition; Processing; Interpretation; Terrain Corrections; Open-Source; Equivalent Sources**

**Abstract.**

In Wallonia, the centre of Hainaut (SW Belgium) is considered the main area where deep geothermal resources are proven. In this region bordering France, the geothermal resource is located within the reservoir formed by the Carboniferous limestones. Productive levels are associated with karst resulting from the dissolution of interbedded massive anhydrites.

To improve current knowledge of deep geological structures and to better delineate the geothermal targets, a new detailed regional gravity anomaly map has been produced. This has involved the integration and harmonisation of legacy and new cross-border gravity data, combined with homogeneous reprocessing of the effect of topography and interpolation using the equivalent source method.The legacy data come from the Belgian and French databases published by the ROB (Belgium) and BRGM (France) respectively, to which unpublished data from the Battaille 1967 campaign have been added. The new

gravity data come from the recent MoreGeo 2019-2022 gravity acquisition campaign. This campaign covers an area of 820 km² with 3,400 gravity stations distributed along densely sampled profiles and displays a mean square error of 39.61 µgal. The multi-scale fusion of non-standardised data was achieved by developing a gravimetric processing workflow based on open-source Python libraries. It includes an advanced drift correction and a terrain correction up to 167 km in extent.

The resulting map is consistent with previous maps, in particular the strong correlation between the anomaly and the

thickness of the Meso-Cenozoic deposits. It also shows new anomalies that may be due to deep-seated structures. It will be used as a basis for modelling the geothermal reservoir, provided that the effect of the Meso-Cenozoic deposits is removed.

**Introduction**

In Belgium, numerous gravity surveys have been carried out for more than a century. These have already been the subject of several reviews, notably in (**Dupont, 2021; Verbeurgt et al., 2019; Everaerts and De Vos, 2012**). These various

acquisition campaigns have made it possible to increase the density of the network of gravimetric stations throughout the country and to refine the mapping of the gravity anomaly at the national level. The most recent mapping is shown in Fig.1 and is taken from (**Everaerts and De Vos 2012**) where the various iterations of the national mapping of the gravity anomaly are presented.



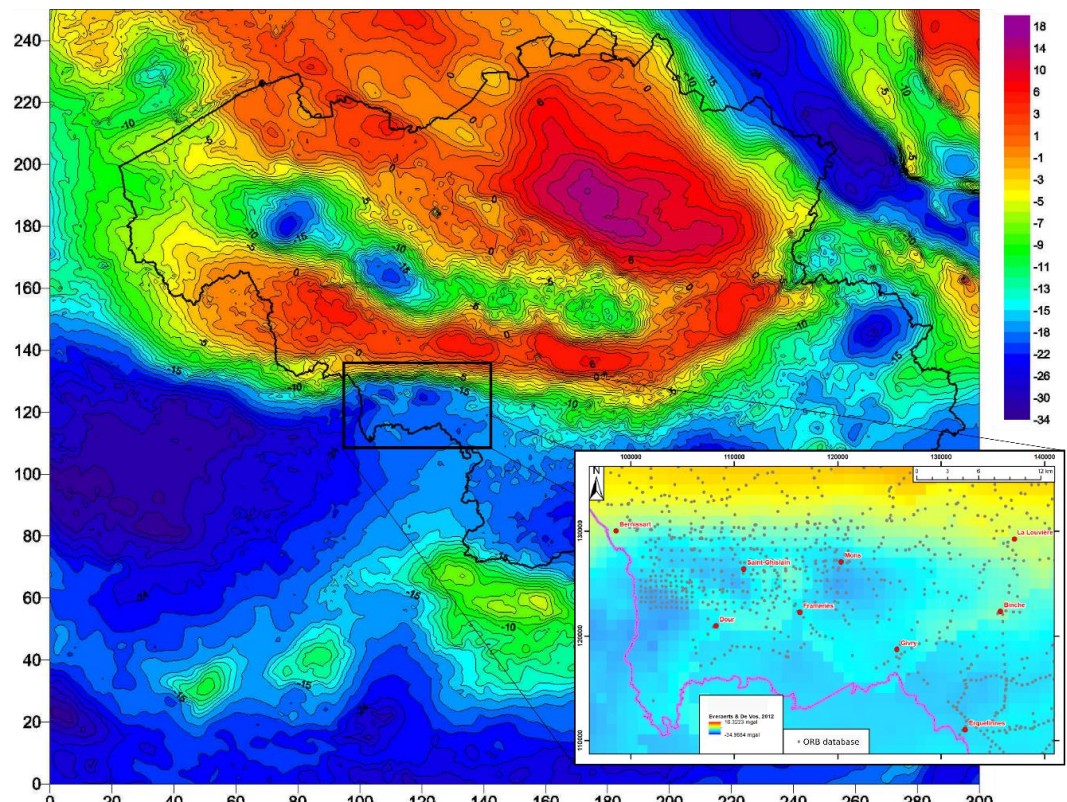

**Figure 1: Map of the Belgian gravity anomaly obtained with a reduction density of 2.67. Focus on the S-W Belgium study area with locations of Belgian gravity stations (Verbeurgt et al., 2019) integrated in the current mapping. Original maps from (Everaerts and De Vos, 2012) and (Dupont, 2021).**

The gravity data used to produce this map are currently available in public databases. These are the gravity database published by the Observatoire Royale de Belgique (ROB) for the Belgian data **(Verbeurgt et al., 2019)** and the Banque Gravimétrique de France for the French data **(BRGM, 2023).** A view of this map in the Mons region, the study area for this work, with the Belgian gravity stations from the ROB database is also shown in Fig. 1.

This last figure highlights the two main (and inextricably linked) shortcomings of gravity anomaly mapping in this area. The first is the resolution of the map. It is currently only available on a 1 km grid **(Dupont, 2021)**, which is not suitable for the sub-regional scale of this work. Secondly, the spatial density of the Belgian stations is rather low and even very heterogeneous, especially in the south of this zone along the French border.

This study area, considered to be the main area where deep geothermal resources are known in Wallonia **(Petitclerc and Vanbrabant, 2011)**, would benefit from in-depth knowledge of the gravity anomaly. On the simplified geological map of the Fig. 2, the carboniferous limestones of the Brabant Parautochton tectonic unit occur in the northern part along a mainly West-East axis. they plunge southwards below thrust sheets : (1) the Haine-Sambre-Meuse overturned thrust sheets, mainly composed by Upper Carboniferous coal measures and (2) the Ardennes Allochton, made by folded Devonian formations, mainly sandstones and shales in this area. The base of the Ardennes allochton is a main thrust fault called «Midi(-Eifel) Fault» in France and Belgium. These tectonic units are covered by younger Meso-Cenozoic formations in the central part of the studied area **(Belanger et al., 2012; Mansy et al., 1999)** (cf. fig. 2).

These Carboniferous limestones have massive anhydrite layers at depth, identified by the Saint-Ghislain deep borehole. Their average density is 3.5 **(Campeol, 2021)**, much higher than that of their limestone host. Some of these deep evaporitic levels show significant karstification and therefore represent the most permeable/productive part of the geothermal reservoir



in the Carboniferous limestones. In the region, the three geothermal plants at Saint-Ghislain, Ghlin and Douvrain currently supply hot water at 65-70°C by exploiting the productive zones of this karstic reservoir, located at depths of between 1,350 and 2,500 m depending on their location.

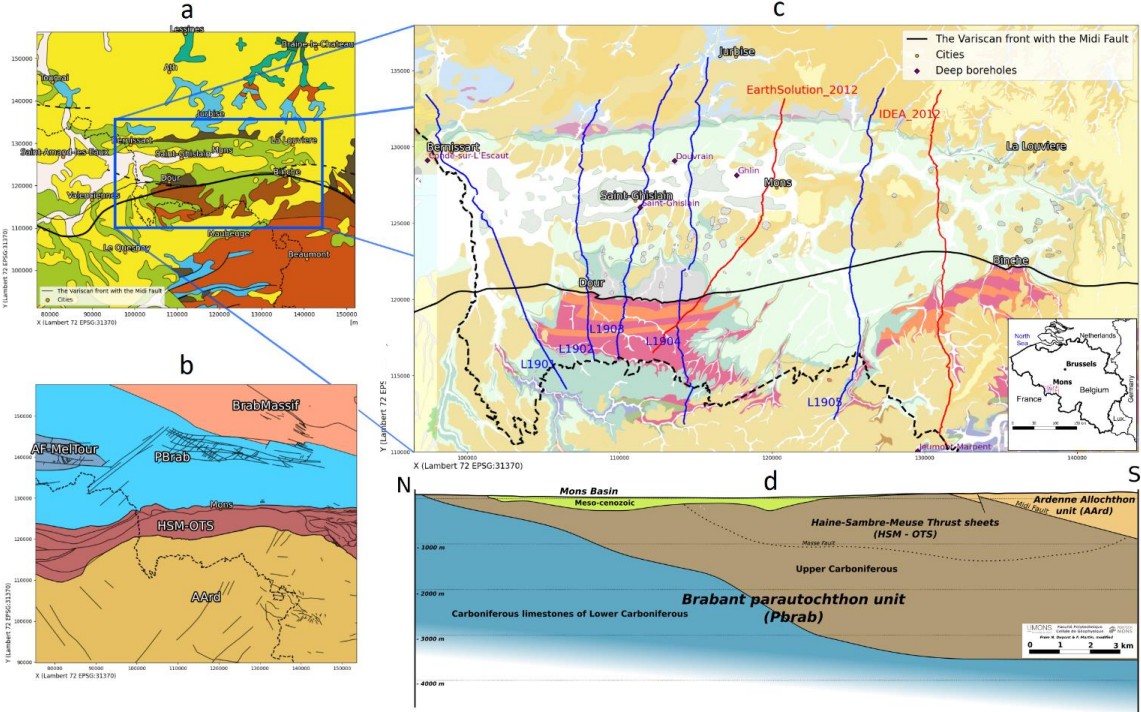

**Figure 2: Simplified geological (a) and structural (b) maps of the study area (S-W Belgium). MBasin : the Mons Basin, Pbrab : the Brabant Parautochthon, HSM-OTS : the Haine-Sambre-Meuse Overturned Thrust sheets, AArd : the Ardenne Allochthon, BrabMassif : the Brabant Massif and the AF-MelTour : the Melantois-Tournaisis faulted anticlinal. Geological map of the Hainaut core area (c) with the various deep boreholes (in purple) and the seismic profiles of the 2DHainaut2012 (in red) and 2DHainaut2019 (in blue) campaigns. Predictive geological section (d) along seismic line L1903, modified from (Dupont, 2021).**

However, the still limited knowledge and the karstic nature of this geothermal reservoir mean that there are major uncertainties regarding the implementation of new geothermal projects. For example, the boreholes at Jeumont and Condé-Sur-l'Escaut in the study area, both of which crossed the Carboniferous limestones of the Brabant Parautochton, did not encounter sufficient hot water productivity for geothermal energy. In fact, at Condé-Sur-l'Escaut, cold water was extracted from thick breccia layers, whereas at Jeumont, the reservoir in question is remarkably thin with few permeable layers, 70   providing a low flow of moderately hot water. This is why, since the 2010s, a series of geophysical studies have been carried out to constrain the geometry of this very specific geothermal reservoir, such as the two seismic reflection campaigns Mons 2012 and Hainaut 2019 **(Dupont, 2021)**.

Recently, gravimetric modelling along the seismic lines of the Mons 2012 campaign located in Fig. 2 was carried out by Dr 75   Michel Everaerts and reported in **(Dupont, 2021)**. He modelled the gravity anomaly that should be observed along these lines based on a density model and a geological model constructed from the interpretation of the seismic sections. The result of this modelling was compared with the gravity anomaly calculated by **(Everaerts and De Vos, 2012)** and gave unsatisfactory results, highlighting the difficulty of reconciling this mapping with consistent density models and/or interpretation of the seismic sections.

As a result, the relevance of the map of **(Everaerts and De Vos, 2012)** in the study area was questioned, mainly because the spatial density of the gravity measurements was insufficient to discriminate between density contrasts **(Dupont, 2021)**. The complexity of the structures identified on the seismic sections would therefore require a refinement of the mapping of the



gravity anomaly, and particularly a densification of the gravity measurements along the seismic lines. In addition to direct modelling along the seismic sections and the construction of a density model, improving the mapping of the anomaly in a sub-regional sense has many other advantages. For example, it would help to better constrain any future modelling of the geothermal reservoir outside the seismic lines. The nature of the reservoir could also be exploited, as its karstified and massive anhydrite zones could appear as negative and positive anomalies respectively, provided that the influence of the Meso-Cenozoic overburden formation is eliminated.

A recent attempt to refine the gravity network in the zone was made by **(Dupont, 2021)**. He used the results of the Bataille 67 and Bataille 68-69 surveys in which the gravity anomaly had been calculated with a reduction density of 2.1. The map obtained is the result of kriging these gravity data on a 250 m grid. The main advantage was the use of an unpublished dataset, the Bataille 67 campaign, which was not included in the ROB database. Unfortunately, only the gravity anomaly values for this dataset were found in the archives of the Belgian Geological Survey. It has therefore not yet been mapped in conjunction with the data used by **(Everaerts and De Vos, 2012)**, mainly because of the different reduction densities used in the corrections.

In order to refine and complete the mapping of the gravity anomaly, it is therefore also necessary to collate all gravity data available in the area. This additional objective involves standardising the corrections applied to the various data sources, in particular the terrain corrections. Therefore, special effort are made to create a modular processing chain using only open-source tools.

**The new MoreGeo 2019-2022 acquisition campaign**

**Equipment used**

Equipment used during the gravimetric campaign included :

- A field micro-gravimeter, the Scintrex CG-6 Autograv™.

According to the manufacturer, Scintrex, and **(Francis, 2021)**, the reading accuracy is 0.1 µgal and the measurements have a standard deviation that can be less than 5 µgal, providing sufficient accuracy for possible future modelling work. The second advantage of this instrument is that it is easily transportable, enabling the acquisition of around forty gravimetric stations per day. Since the aim of these campaigns was to achieve accuracy in the order of tens of µgal while maintaining maximum mobility in the field, accuracy and mobility are the two most important parameters.

- A GNSS antenna (RTK option), the SXblue Premier RTK II
- A field controller for GPS systems, the SXPad 1000p
- A carbon fibre rod for the GPS antenna

The field controller establishes a Bluetooth connection with the GNSS antenna and a connection to the WALCORS GNSS network via the cellular network. This enables RTK corrections (via the NTRIP protocol) to achieve horizontal and vertical accuracies of less than one centimetre.

**Design of the new surveys**

The design decisions for this gravimetric campaign consist in defining the location of the stations and their distribution over the surface of the study area. In general, the location of the gravity stations follows the seismic lines of the Mons2012 and Hainaut2019 campaigns presented above. The topographic survey of the stations was carried out with centimetre accuracy thanks to the corrections made by the Walcors network. Latitude, longitude and altitude were measured in the WGS84 coordinate system (EPSG:4326). A distance of 50 m was chosen as the distance between the stations. This is because it is the same distance observed between the vibration points of the two seismic acquisition campaigns, and it is desirable to respect a certain consistency in the spatial sampling used by these two geophysical methods. Indeed, one of the specific objectives of this MoreGeo campaign is to characterise more precisely the gravity anomaly along the seismic lines of these two campaigns. In this way it would be possible to couple gravimetry with seismic reflection data in advanced processing and imaging techniques such as constrained inversions.

As far as the acquisition parameters are choosen, the most important thing is to define the number and duration of the



acquisition cycles carried out at each station. For MoreGeo 2019-2022, these have been defined as three consecutive measurement cycles per station, with a recording duration of 30 seconds for each cycle. This means that each record per station lasts 90 seconds. At the end of each cycle, the gravimeter calculates the mean, standard deviation, and standard error
over a set of 30 measurements. The acceptance criterion for the gravimetric record of a station is defined by the maximum difference between two of the three measurements, which must be less than 10 µgal. If this is not the case, the measurement must be repeated immediately.

Other acquisition parameters have been set, such as the size of the level windows at 10 arcseconds, a parameter that controls the horizontality of the instrument, and the limit for the inclination of the gravimeter. It depends on the accuracy expected
by the operator and, in principle, 10 arcseconds corresponds to a measurement error of the order of 1 µgal, while it increases to 10 µgal for 20 arcseconds. The delay before starting a measurement has been set at 5 seconds to allow the operator to move away and limit the influence of his presence on the gravimeter's measurement.

In order to assess the instrumental drift of the gravimeter, a phenomenon inherent to this type of acquisition, acquisition loops and base stations were also included in the design of this campaign. This instrumental drift, studied in particular by
**(Francis, 2021)**, is related to the coupled effects of the constant elongation of the quartz spring of the Scintrex CG-6 gravimeter and the compensation that the latter constantly evaluates. By taking repeated measurements of a station at different time intervals, it is possible to assess the instrumental drift after eliminating other temporal effects such as the influence of lunar-solar tides. As part of the MoreGeo 2019-2022 campaign, closing loops have been integrated into the acquisition procedure. These are slightly different between 2019 and 2022 and consist, for example, of:

- Daily measurements from fixed stations in rue du Joncquois in Mons at the beginning (2019 and 2022) and the end of the day (2022);
       - The measurement of a second fixed base specific to each profile (2019);
       - The first measurement of the day in the field at a station that has already been measured in previous days, then called an intra-profile loop (2022);
- The measurement of stations in other lines located at the intersections of the profile surveyed, then called the inter-profile loop (2022);
       - Resampled stations (end of 2022).

With regard to this last point, 3% of all stations (i.e. 120 stations) were resampled in October 2022 in order to assess the quality of all the measurements taken as part of this MoreGeo 2019-2022 campaign and to validate the work carried out on
drift correction. These stations were selected semi-randomly to ensure a good distribution of points along the lines.

**Location**

The 2019 gravity surveys follow the lines of the Hainaut2019 seismic campaign. They cover the Cœur du Hainaut area with 5 lines of about 20 km in length, oriented North-South, evenly distributed over the survey area. A total of 1,726 stations were recorded, giving a cumulative profile length of approximately 86 km. The location of the 2019 lines is shown in Fig. 3
alongside those of the 2022 campaign. The table below provides a summary by profile number.

A major constraint in preparing the lines was to avoid major roads and densely populated areas. This not only saved time by reducing administrative permitting procedures, but also reduced sources of vibration, resulting in better quality, low dispersion acquisitions.



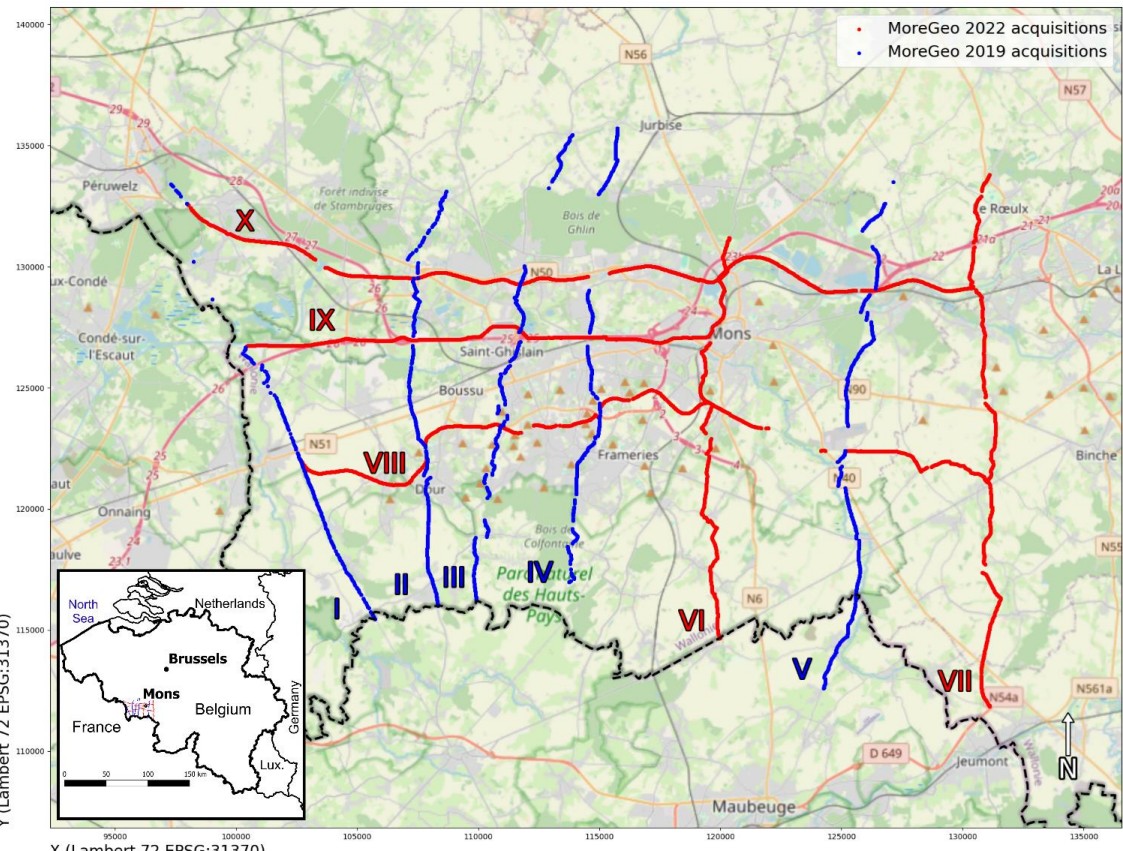

**Figure 3 : Lines from the MoreGeo 2019-2022 gravity survey, with the 2019 acquisition lines in blue and the 2022 acquisition lines in red. ©OpenStreetMap Distributed under the Open Data Commons Open Database License (ODbL) v1.0.**

In 2022, the second phase of the MoreGeo 2019-2022 gravity campaign was launched in the Hainaut region with the aim of collecting a set of data to complement the 2019 surveys. This new acquisition campaign lasted about 6 months and included two lines running north-south along the seismic lines of the Mons2012 campaign and three lines running east-west. A total of 2456 stations were acquired, representing approximately 123 km of profile. The table below provides a summary by profile number.

For similar reasons to 2019 (minimising sources of vibration and regulatory approval processes), the routes have been carefully chosen to avoid major roads, densely populated areas and private land. As a result, almost all lines are located along RAVeLs (autonomous network of non-motorised paths), towpaths and little used paths. Figure 3 shows the lines acquired in 2019 (in blue) and those acquired in 2022 (in red). However, three areas along the seismic lines could not be covered by the gravimetric survey. As can be seen on this map, these are densely forested areas where the accuracy of the topographic surveys could not be achieved, making it impossible to acquire data.

**Data formatting**

Following the acquisition campaign, pre-processing operations were performed on the measured data ($g_{Measured}$). These included standardising the data structures and metadata used between the 2019 and 2022 surveys, and merging the GNSS and gravity data. Following this pre-processing, various corrections were applied to generate the Bouguer anomaly map from the gravity measurements and to study only the information relating to the distribution of geological masses in the Hainaut subsoil. This makes it possible, for example, to eliminate the effect of relief, latitude and altitude variations between measurements, the effect of tides, instrument drift, etc. In the rest of this document, corrections represent the



removal of these effects. Therefore, when a phenomenon has a negative effect on the Bouguer anomaly measurements, its correction is expressed with positive values. These were implemented using notebooks written in Python 3. A Python module containing all the functions developed and used in these processes was also created.

**Earth tide correction ($C_{Tide}$)**

First, the temporal effects that affect gravimetric measurements must be corrected. The reading taken by the gravimeter at a
gravity station varies with time. This is partly due to the natural phenomenon of the lunisolar tide, which causes a variation in g that is perfectly measurable at any point on the Earth at any time. In practice, there are models for calculating this effect as a function of the latitude and longitude of the measuring station and of time. The model used in this paper is from **(Longman, 1959)**.

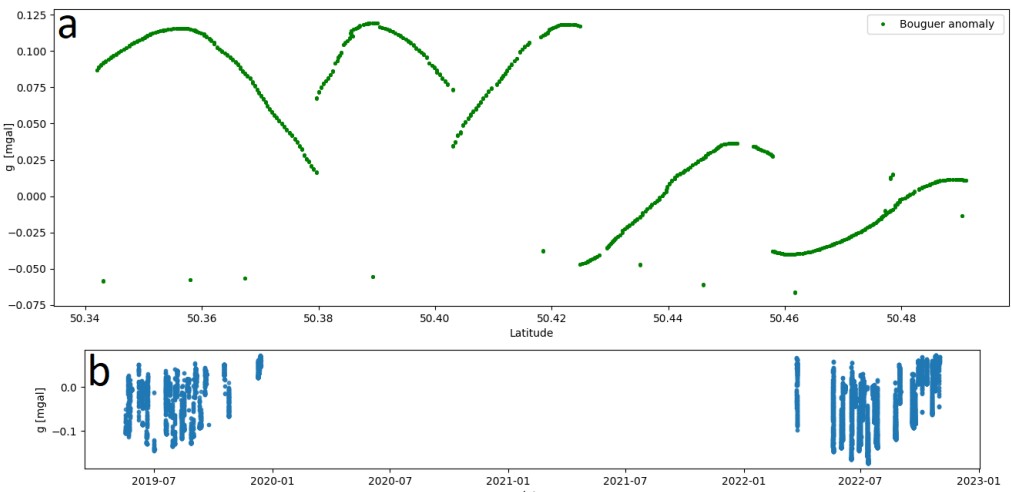

**Figure 4. (a): Effect of the lunisolar tide correction on the processed gravity data from line 6, S - N. (b): Effect of lunisolar tides**
**over the entire MoreGeo 2019-2022 period.**

Figure 4 shows the range of this correction, which is between -0.2 and 0.1 mgal and highlights the periodic pattern of the correction, which effectively demonstrates the cyclical aspect of the phenomenon. In particular, the Fig. 4.a, which shows the profile 2 recordings made over a short period, highlights the daily effect of the moon. This panel also illustrates the 7-day period required to produce this profile. The isolated dots represent the resampling stations for this profile, which were
measured much later. Figure 4.b shows the low-frequency effect of the Sun on this tidal phenomenon.

**Drift correction ($C_{Drift\ intra-period}$ et $C_{Drift\ inter-period}$)**

As discussed above, it is quite common during a gravimetric survey to observe instrumental drift related to the technology of the measuring instrument. For example, in the case of superconducting gravimeters studied over a very long period, the instrumental drift of these instruments can be approximated by a linear or quadratic function over time **(Van Camp and**
**Francis, 2007)**, provided that the acquisition period does not exceed 10 years. Beyond that, an exponential function is better suited to describe the drift phenomenon.

In the case of this MoreGeo 2019-2022 campaign, the technology used by the Scintrex CG-6 relative gravimeter is a fused quartz spring system with electrostatic cancellation **(Phillips, 2015; Francis, 2021)**. With this type of sensor, the instrumental drift, for example, is related to an effect of the elongation of the gravimeter spring over time. According to
**(Francis, 2021)**, the drift of this instrument is quadratic and its growth decreases with time. Given the duration of the MoreGeo campaign and the literature, it is therefore likely that the drift found here can be approximated by this type of function over time.

In terms of magnitude, the manufacturer Scintrex estimates the range of daily drift to be between 0.02 and 0.2 mgal per day. The more recent study by **(Francis, 2021)** estimates the drift of the Scintrex CG-6 to be a few tens of μgal per hour.




Therefore, it seems necessary to estimate and correct the instrumental drift encountered during the MoreGeo2019-2022 campaign using the closure loops presented in the acquisition campaign section. The instrumental drift, initially estimated using only the campaign base stations, is shown in fig. 5:

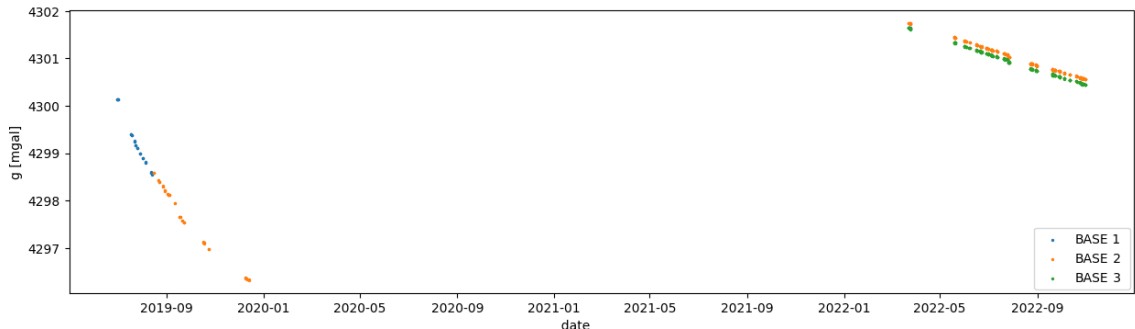

**Figure 5: Estimated instrumental drift between 2019 and 2022 on the three campaign bases.**

**Principle of the method**

The real evolution of the drift can be estimated using closed loops made up of stations visited several times, either systematically or occasionally. All these stations, visited successively at different times, form loops that can be used to approximate the real drift of the instrument. It is therefore a good idea to use all the information available, and not just the measurements taken at the campaign bases, to approximate the drift. In fact, some of the campaign bases were not used
throughout the MoreGeo campaign, such as the first one, which was only surveyed in 2022, and some lines (notably the line 5) were surveyed without systematically returning to the bases.

Each loop is therefore made up of a pair of recordings (separated by at least 5 minutes) made at the same station, making it possible to approximate the linear variation of g with time ($\Delta g/\Delta t$) that occurs over a time interval ($\Delta t$). To determine the drift within a loop, the measurement retained from the three recordings for each of the two acquisitions is the one with the
lowest standard error. The value of the drift variation is assigned to the centre of the time interval framed by the loop (approximated by the secant), since it is hypothesised that the drift follows an at most quadratic law. In fact, the slope of the tangent of a quadratic function at the centre of an interval corresponds to the slope of the secant in that interval.

$$g(t) \equiv y = at^2 + bt + c \qquad (1)$$

$$m_{\sec ant} = \frac{y_2 - y_1}{t_2 - t_1} = \frac{a(t_2^2 - t_1^2) + b(t_2 - t_1)}{t_2 - t_1} = a(t_2 + t_1) + b \qquad (2)$$

$$m_{center} = g'\left(\frac{t_1 + t_2}{2}\right) = 2a\left(\frac{t_1 + t_2}{2}\right) + b = a(t_2 + t_1) + b \qquad (3)$$

From the point of view of statistics and uncertainty calculation, the average $g^*$ of the measurements $g_i$ made on n (at least 30) recordings of g in the field follows a normal distribution of the form

$$\frac{\sum_1^n g_i}{n} = g^* \to N\left(g, \frac{\sigma}{\sqrt{n}}\right) = N(g, \varepsilon) \qquad (4)$$

Since there is no uncertainty about the time of measurement, the drift also follows a normal distribution such that :

$$\left(\frac{\Delta g}{\Delta t}\right)^* = \frac{g_2 - g_1}{\Delta t} \to N\left(\frac{\Delta g}{\Delta t}, \frac{\sigma_1 + \sigma_2}{\Delta t \sqrt{n}}\right) = N\left(\frac{\Delta g}{\Delta t}, \frac{\varepsilon_1 + \varepsilon_2}{\Delta t}\right) \qquad (5)$$

Since the true standard deviation of the distribution is unknown, the standard deviations associated with the two measurements, which are unbiased estimates of the variance, follow a Student distribution of the form :





$$s = \sqrt{\frac{1}{n-1}\sum_{i}^{n}(g_i - g^*)^2} \rightarrow \tau_{(n-1)} \qquad (6)$$

Thus, the error in estimating the drift in this loop, denoted ε, is a function of the sum of the standard errors of the two
measurements used to calculate the drift and the Student's t coefficient for a given 1-α confidence level:

$$\left[ g^* - t_{\left(\frac{\alpha}{2}\right)}^{(n-1)} \varepsilon; \; g^* + t_{\left(\frac{\alpha}{2}\right)}^{(n-1)} \varepsilon \right] \qquad (7)$$

$$\left[ \frac{\Delta g - t_{\left(\frac{\alpha}{2}\right)}^{(n-1)} (\varepsilon_1 + \varepsilon_2)}{\Delta t}; \; \frac{\Delta g + t_{\left(\frac{\alpha}{2}\right)}^{(n-1)} (\varepsilon_1 + \varepsilon_2)}{\Delta t} \right] \qquad (8)$$

$$\left[ \frac{\Delta g}{\Delta t} - \frac{t_{\left(\frac{\alpha}{2}\right)}^{(n-1)} (\varepsilon_1 + \varepsilon_2)}{\Delta t}; \; \frac{\Delta g}{\Delta t} + \frac{t_{\left(\frac{\alpha}{2}\right)}^{(n-1)} (\varepsilon_1 + \varepsilon_2)}{\Delta t} \right] \qquad (9)$$

The pairs are used to construct a graph of the variation of g over time ($\Delta g/\Delta t$) as a function of time (t) (see Fig. 6). Next,
pairs with drift above a threshold (typically 1 mgal/day) are removed because they do not represent a plausible measure of
instrument drift, but rather the effect of an external disturbance. For example, a loud noise such as a passing vehicle during
one of the measurements of the couple, a change in the height of the tripod, incorrect levelling of the gravimeter (horizontal
levelling). It was also necessary to filter for the maximum duration and position of the time interval between measurements
within the same pair.

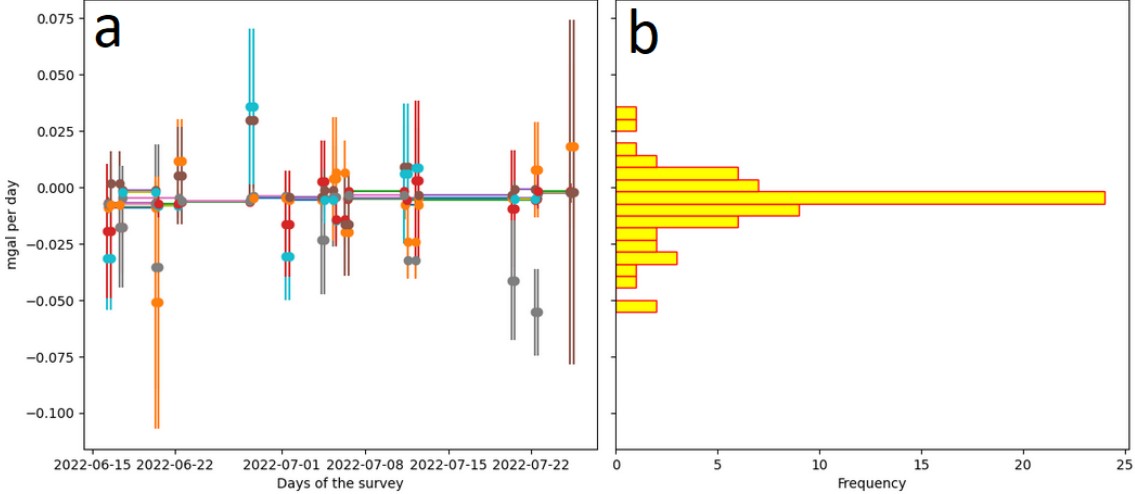


**Figure 6: Evolution of the drift over time between June and July 2022 with the measured loops and associated error (a) and their
histogram (b).**

The loops filtered in this way are used to approximate the instrument drift using Weighted Least Squares (WLS), which is a
linear least squares interpolation that weights the drift data by its uncertainty. This WLS is also performed within a sliding
window. In other words, at each pair centre, a WLS of a chosen degree (1 or 0) is performed over a limited time window
and its value at that time is assigned to the drift ($\Delta g/\Delta t$). In this way, the drift at each point is evaluated by weighting only
the drifts observed around that point.

The parameters used in the sliding WLS are therefore the width of the time window and the degree of the polynomial used
in the WLS. The possible values for each of these parameters are, for the width of the time window, between 1 day and the





maximum recorded time deviation and, for the degree of the polynomial, between 0 and 1. The values adopted are the optimum values among those allowed which, after correction for drift, give the least scatter of the gravimetric measurements at the three bases of the campaign.

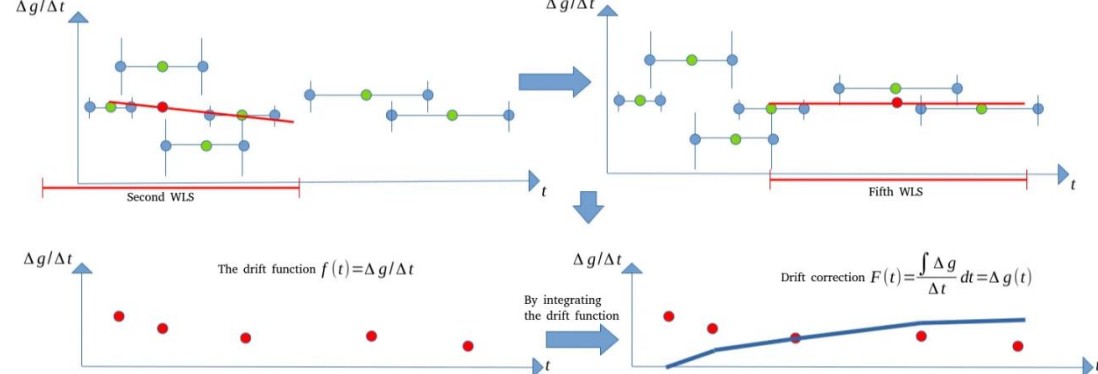

Figure 7: Illustration of the principle used to determine the drift function using Moving Weighted Least Squares (MWLS). The blue points are the two measurements associated with a closing loop represented by the green points. The red points are
the result of the MWLS calculated from the green points.

Next, the curve connecting the observed drifts (represented by the points in red in Fig. 7) is called the drift function. This is integrated over time (in blue in Fig. 7) to give the drift correction to be applied to the gravimetric measurements from a reference date. This drift correction therefore varies linearly or quadratically with time.

$$\int_{t_0}^{t_1} \frac{\Delta g}{\Delta t} \mathrm{d}t = \Delta g(t) \tag{10}$$

It is not necessary to filter pairs with a very small time interval because the error in the drift estimate (*eps*) becomes very large and these doublets therefore have little weight in the WLS.

**Application of the drift correction**

In the MoreGeo gravity campaign, the total duration of the campaign was such that it was not possible to approximate the drift using a single drift function. The result of this attempt is shown in the Fig. 8.a. This is due to the shape of the long-term
drift and its decay, which could not be modelled by a single polynomial. This observation can be explained by the fact that the gravimeter was returned to the manufacturer in 2020, which probably influenced the shape of the drift between the two phases of the campaign (2019 and 2022).

It was therefore decided to determine a drift function within each phase of the MoreGeo campaign, i.e. one for 2019 and one for 2022. The result is shown in the Fig. 8.b. Although it's much better than the previous one, unfortunately it still does not
meet the requirements set at the time of this treatment. In fact, the dispersion of the gravity measurements at the three bases within these two periods still exceeds the accuracy target set at 10 µgal.



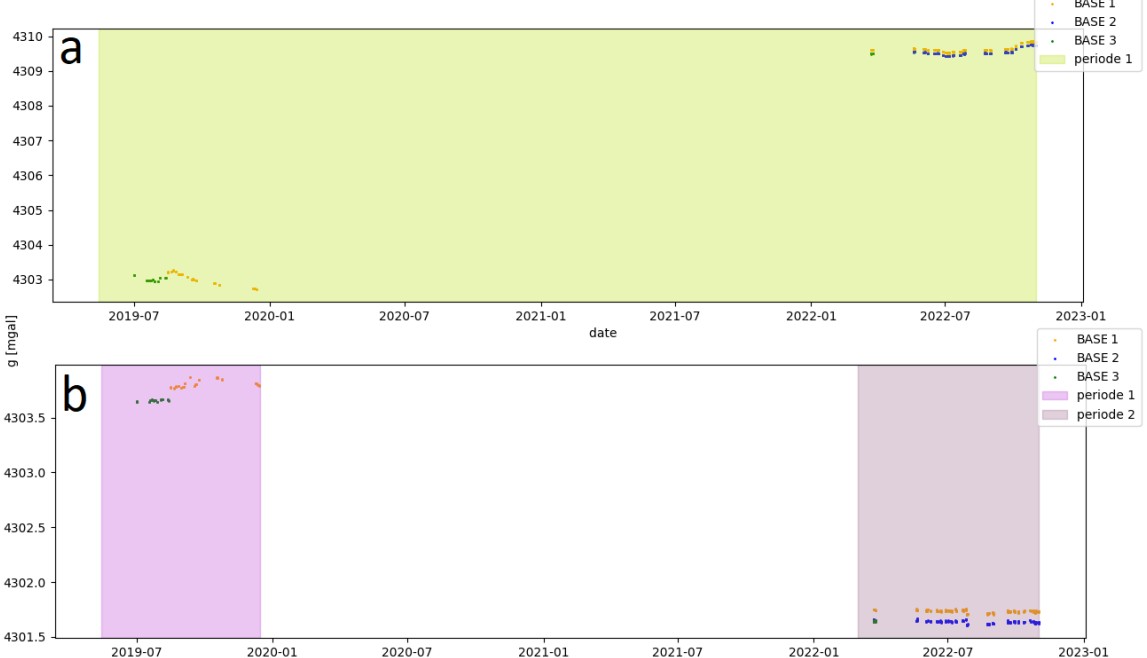

**Figure 8: Result of intra-period drift correction after determining a one-period (a) and two-period (b) drift function.**

Nevertheless, it turns out that dividing the 2019-2022 MoreGeo campaign into periods in order to determine a single drift function within these periods significantly improves the result. However, it is clear that dividing the MoreGeo campaign into an excessive number of periods would lead to an ever finer dispersion but a completely meaningless result. It is therefore necessary to determine the optimum number of periods where the number of parameters to be included in the drift model is most relevant in relation to the precision obtained. For this purpose, the Akaike Information Criterion (AIC) and its derived form, the Bayesian Information Criterion (BIC), have been evaluated. These criteria are defined with the eq. (11) and 12 respectively.

$$AIC = 2k - 2\ln(L) \qquad (11)$$

$$BIC = \ln(n)\,k - 2\ln(L) \qquad (12)$$

Where $k$ is the number of parameters (and therefore periods) to be included in the model, $n$ is the number of observations and $L$ is the objective function, which in this case is a 10 µgal dispersion of the gravimetric measurements of the three bases within the periods after correction for drift. The main difference between these two criteria is the weight given to the number of parameters, which in the case of BIC depends on the number of observations. The result of these two criteria is shown in Fig. 9 and shows that the optimal number of periods to include in this drift correction step is 5. These criteria also highlight the significant gain in accuracy between the drift function calculated over the whole campaign and two functions calculated within each sub-campaign.

The upper and lower limits of the periods used in these tests include only precise moments that separate important phases of the acquisition, such as the acquisition of different lines, and that allow the drift to be corrected as much as possible within these periods. Based on this observation, the MoreGeo 2019-2022 campaign was therefore divided into 5 periods. For each of these periods, an intra-period drift correction was calculated according to the principle presented above. The result of this intra-period drift correction is also shown in Fig. 9.





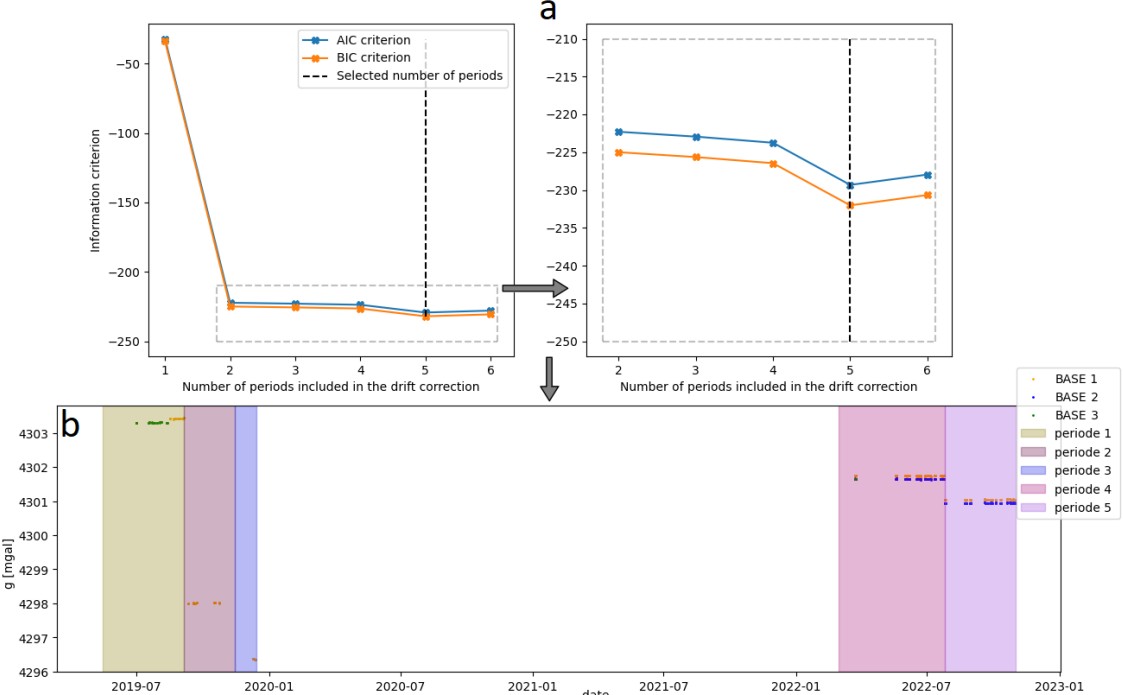

**Figure 9: Selection of the number of periods according to the AIC and BIC criteria (a). Associated result of the intra-period drift correction on the three campaign bases (b).**

Following this division into periods and the intra-period drift correction, the gravity measurements are now corrected in relation to the start date of their respective periods. We now need to correct for the drift over the whole campaign, known as the inter-period correction, based on gravity measurements taken at the same station between different periods. A first simple approach would be to determine the shifts to be applied between periods based on the average measurements obtained per period on one or more reference bases. However, relying solely on the measurements made at the reference points is probably not the most effective way of repositioning the different periods with respect to a reference period. Furthermore, as already explained, there is no single baseline used throughout the MoreGeo 2019-2022 campaign, as three campaign baselines and five profile baselines were used in succession. Finally, there is the question of the choice of the reference period on which the whole campaign is based and on which the offset relationships will depend.

To respond to these different problems, all the closing loops that connect two acquisitions of the same station in two different periods have been selected. These 111 loops are used to introduce the offset relationships necessary to reposition the periods. Most of the inter-period shift relationships are almost unambiguous, as the different relationships linking these periods determine very similar shift values. On the other hand, other pairs of periods have shift relationships whose values appear to be very scattered. It therefore seems necessary to filter these relationships between periods beforehand. A first filter was applied when the dispersion of the shifts for the same pair of periods is greater than a critical threshold (the threshold is set at a standard deviation of 100 µgal). In this situation, all offset relationships for that pair of periods are removed from the problem because this extreme dispersion is considered too large and invalidates all of these offset relationships. The second filter is applied when the scatter of the offsets is greater than a tolerance threshold (a function of the desired accuracy during the campaign, in this case 10 µgal) for the same pair of periods, in which case the extreme values are removed until the scatter of the offset relationships is less than the desired accuracy. The Fig. 10.a shows the 80 offset relationships retained for period repositioning after applying these two filters.



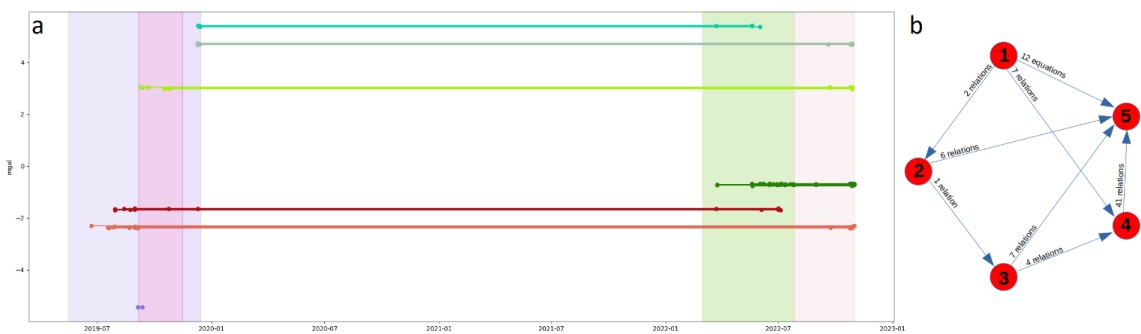

**Figure 10: Inter-period shift relationships after intra-period drift correction and filtering (a). Graph of inter-period shift relationships (b).**

After filtering, the various shift relations form an overdetermined system of the form:

$$\begin{bmatrix} d_{\{15\}} \\ d_{\{15\}} \\ d_{\{15\}} \\ d_{\{23\}} \\ d_{\{23\}} \\ ... \\ d_{\{45\}} \end{bmatrix} = \begin{bmatrix} S_{\{5_1\}} - S_{\{1_1\}} \\ S_{\{5_2\}} - S_{\{1_2\}} \\ S_{\{5_3\}} - S_{\{1_3\}} \\ S_{\{3_1\}} - S_{\{2_1\}} \\ S_{\{3_2\}} - S_{\{2_2\}} \\ ... \\ S_{\{5_x\}} - S_{\{4_x\}} \end{bmatrix} \tag{13}$$

where $S_{i_x}$ are the averages of the station x acquisitions made in period i and $d_{ij}$ are the unknown offsets between the two periods i and j for the same pair of x acquisitions. Figure 10 shows the graph of the relationships between periods. According to graph theory, this is an incomplete graph because there are not necessarily links between each node (in this case the periods). In addition to these equations, we also need to include closure relationships, i.e. equations that loop over the same period and use more than two different offsets. Given the graph above, the closure relations, determined using a graph traversal algorithm, are:

$$\begin{bmatrix} d_{\{23\}} + d_{\{35\}} - d_{\{25\}} \\ d_{\{14\}} + d_{\{45\}} - d_{\{15\}} \\ d_{\{12\}} + d_{\{25\}} - d_{\{15\}} \\ d_{\{34\}} + d_{\{45\}} - d_{\{35\}} \end{bmatrix} = \begin{bmatrix} 0 \\ 0 \\ 0 \\ 0 \end{bmatrix} \tag{14}$$

All these 84 equations form an overdetermined system with 8 unknowns, the $d_{ij}$. The method used to solve this system is then based on the Moore-Penrose inverse matrix, which determines the best solutions of the system in the sense of least squares **(Barata, 2012; MacAusland, 2014)**:

$$Ax = b \tag{15}$$

$$A^*Ax = A^*b \tag{16}$$

$$x = (A^*A)^{-1}A^*b \tag{17}$$

With A*, the adjoint matrix of A. By introducing the Moore-Penrose inverse matrix A⁺:

$$A^+ = (A^*A)^{-1}A^* \tag{18}$$

$$x = A^+b \tag{19}$$

Once this system has been resolved, the offsets to be applied between periods are now known, as shown in Table 1.

| Periods | 1 | 2 | 3 | 4 | 5 |
|---|---|---|---|---|---|
| 1 | - | d12 | / | d14 | d15 |




| Periods | 1 | 2 | 3 | 4 | 5 |
|---|---|---|---|---|---|
| 2 | d21 | - | d23 | / | d25 |
| 3 | / | d32 | - | d34 | d35 |
| 4 | d41 | / | d43 | - | d45 |
| 5 | d51 | d52 | d53 | d54 | - |

Table 1 – Table of known inter-period shift relationships.

We now need to determine a reference period to which all the others can be repositioned. In the case of MoreGeo, there is one period, 5, to which all the others can be repositioned directly using the current solutions of the system of equations. However, this ideal case does not occur systematically, and an algorithm has been developed to complete this table

beforehand, so that all periods can be repositioned to any other reference period.

This algorithm iterates through the rows and columns of the table. If the offset $d_{ij}$ between periods i and j is unknown, the algorithm recovers all existing pairs ($d_{ik}$, $d_{kj}$) (for k possible periods). These pairs form the various shortest paths connecting two periods that do not have a direct offset. Their average is then assigned to the offset $d_{ij}$ in the table. This algorithm is applied until the table is filled. In other words, $d_{ij}$ is determined by eq. (20):

$$d_{ij} = \frac{1}{n} \sum_{k}^{n} \left( d_{ik} + d_{kj} \right) \text{ with } k \subset [1{:}\text{nbr of periods}]/(i, j) \tag{20}$$

As the previous system of equations has been solved in the sense of least squares, the offsets indicated in the completed table depend on the period chosen and have a slight influence on the repositioning result. In the end, the reference period chosen is the one that includes the measurements at the three bases of the campaign with the least dispersion. In the case of MoreGeo, this is the fifth, which includes the resampling period at the end of October 2022. The dispersions, symbolised by

the standard deviation of the measurements taken at each of the three bases during the MoreGeo 2019-2022 campaign, are 9.6, 6.8 and 7.9 µgal, which is below the target accuracy of 10 µgal (Fig. 11.a). The combined effect of inter- and intra-periodic corrections can also be seen in the gravity measurements for line 2 shown in the Fig. 11.b.

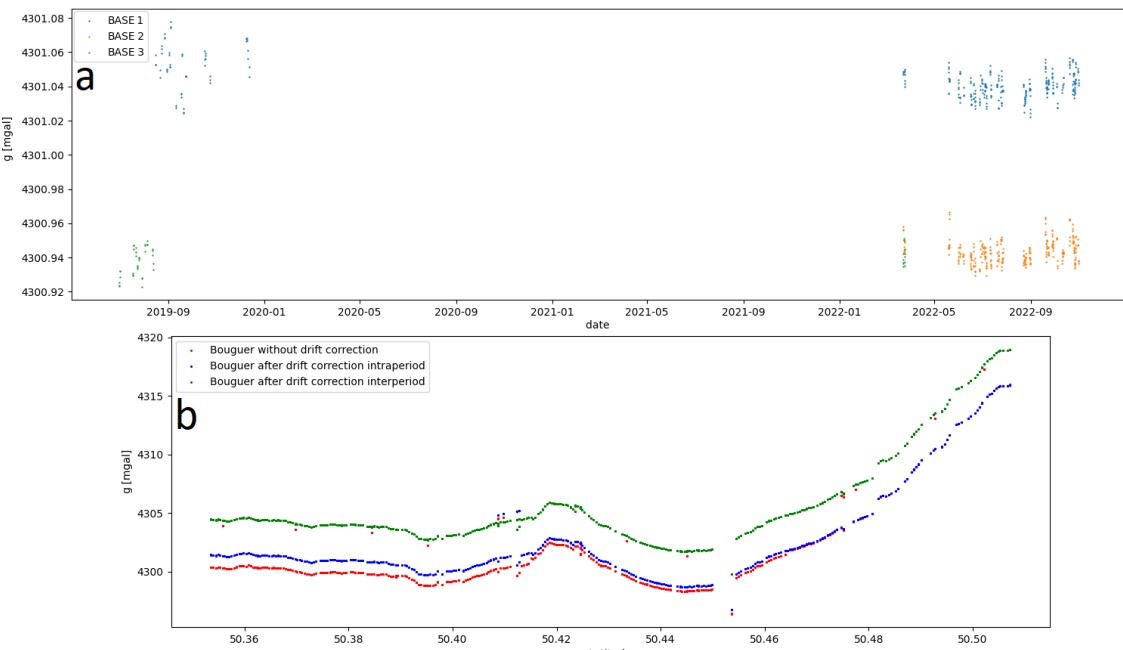

**Figure 11: All gravity measurements taken on the three bases of the 2019-2022 MoreGeo campaign, after repositioning the**
**periods to the fifth reference period (inter-period drift correction) (a). Effect of drift correction on processed gravity data, Profile**



**2, S - N. The raw measurements (purple), Bouguer anomaly with (green) and without drift correction (red) are shown (b).**

## Complete Bouguer correction

As explained by Bullard in 1936 **(Bullard, 1936)** and summarised in **(Nowell, 1999)**, variations in altitude and topography affect the gravitational attraction. The corrections associated with these elements consist of three distinct effects. Firstly,
differences in altitude between stations affect the measurement of g because the distance between the stations and the Earth's centre of mass is different. To compare gravity measurements without this effect, two corrections are regularly applied. The first, the free air correction (Bullard A), is used to bring the heights of the measuring stations down to a reference height defined with respect to the reference datum. The second, the simple Bouguer correction (Bullard B), eliminates the influence of masses located between the height of the station and the reference height. The sum of these two
corrections is called the altitude correction:

$$\text{Altitude correction} = \text{Free-air correction} + \text{Simple Bouguer correction} \qquad (21)$$

The topography of the land around a station also affects the measurement. The presence of valleys, hills, dumps and other features all affect the direction and value of the gravity acceleration vector as they represent mass (or mass deficit) in the vicinity of the station. In most cases, the topographic effect is negative, since the presence of these topographic features
reduces the norm of the gravity acceleration vector, as opposed to the simple Bouguer effect, which increases it. The most common exception to this rule is when the topographic correction surface is below the reference height, for example in bathymetry. The topography is then considered by the topographic correction (Bullard C). This topographic correction is often combined with the simple Bouguer correction to form the terrain correction:

$$\text{Terrain correction} = \text{Topographic correction} + \text{Simple Bouguer correction} \qquad (22)$$

Finally, the complete Bouguer correction is the result of adding the three components:

$$\text{Complete Bouguer correction} = \text{Free-air correction} + \text{Terrain correction} \qquad (23)$$

## Free-air correction ($C_{\text{Free air}}$)

The outdoor correction reduces the measurements to a common height but ignores any excess or deficit of mass with respect to the chosen reference level. To eliminate the influence of the distance from the Earth's centre on the altitude, **(Heiskanen**
**and Moritz, 1967)** proposed the following formula, which has since been adopted by the International Union of Geodesy and Geophysics (IUGG):

$$\text{Free-air correction} = -(0.308769097 - 0.000439773125 * \sin^2(\phi)) * h + 0.0000000721251838 * h^2 \qquad (24)$$

This formula includes h and φ, which represent the altitude and latitude of the stations respectively. In the case of MoreGeo, altitude is measured with respect to the GRS80 ellipsoid in the WGS84 geodetic datum. Figure 12 shows the application of
this correction to profile 7 data. After analysing the results, we can see that the correction values are significant, up to 60 mgal in absolute value. Figure 12 also shows that the shape of the outdoor correction is quite opposite to the shape of the profile's altimeter curve.

When the height of the geoid varies rapidly over the scale of a gravity survey, as is the case in mountainous regions, this causes multimetric differences between the height of the geoid and that of the ellipsoid and generates indirect effects
**(Talwani, 1998)** that can affect the calculation of free-air and simple Bouguer corrections. In the case of the MoreGeo campaign, this effect is negligible given the topography encountered and the small variation in the height of the geoid compared to that of the ellipsoid.



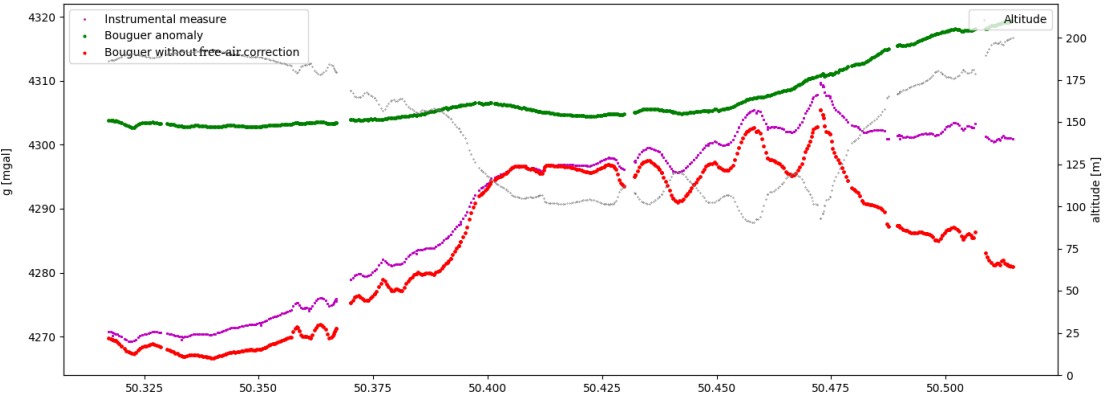

**Figure 12 - Effect of the free-air correction on the data from Profile 7, S - N. Plotted are the topographic survey (grey), the raw measurements (purple), the Bouguer anomaly with (green) and without free air correction (red).**

**Terrain correction**

**Principle of the method**

As mentioned above, it is necessary to correct for the effects of topography to account for the mass deficits and surpluses associated with the relief around gravity stations, which are not corrected by the simple Bouger correction. Introduced by **(Hayford and Bowie, 1912)**, the use of topographic corrections became widespread thanks to the work of **(Hammer, 1939)**, who introduced the method in graphical form using abacuses **(Martelet and Debeglia, 2001)**.

Unfortunately, these graphical methods are tedious and time consuming, especially in the context of this work, given the amount of data used and the extent of the region to be considered in the correction. These methods require manual calculations for each of the stations, and the topographic extent considered for this correction is generally 167 km around the stations. This 167 km extension, proposed by **(Bullard, 1936)** as a reference extension, represents the optimum radius to be used in topographic corrections in order to minimise the effects associated with the spherical shape of the Earth, as can be seen in Fig. 13. Taken from **(Martelet and Debeglia, 2001)**, it models this effect, known as the "Bullard effect", as a function of the altitude of the gravity station and for different extents considered in the correction, and shows that the extent for which the effect of the spherical cap is minimal, whatever the altitude of the station, is 167 km. In addition, the proximity of the measurements to France, where the topographic corrections were extended to the whole country in 2002 in **(Martelet et al., 2002)**, also justifies this choice.

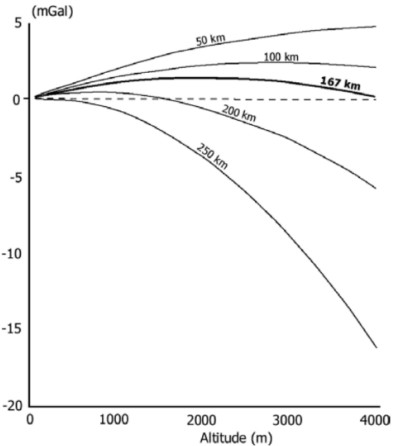



**Figure 13: Bullard effect as a function of station altitude (0 to 4000 m) modelled for the different extents considered in this correction. Taken from (Martelet anf Debeglia, 2001), based on (LaFehr, 1991).**

Numerical methods exist, including the one introduced by **(Nagy, 1966a)**, which is based on the calculation of the influence of rectangular prisms on the vertical component of the gravitational field at an observation point. This method, which is easy to implement numerically **(Nagy, 1966b)**, has also been summarised by simplifying the expressions for the gravitational potential of a rectangular prism and its derivatives in **(Nagy et al., 2000; Nagy et al., 2002)**. Other methods exist, such as those based on the calculation of the gravitational potential of tesseroids and mass lines **(Heiskanen and Moritz, 1967; Heck and Seitz, 2007; Makhloof, 2008)**. The first, which is more time-consuming and resource-intensive to compute than the prism method, allows the curvature of the Earth to be optimally taken into account in large surveys and is particularly useful when there are very large level differences between stations. The second, using lines of mass, is often used to simplify calculations when distances become significant. A comparison with the prism method can be found in **(Heck and Seitz, 2007)**. These different methods can also be coupled and used as a function of distance from the station to optimise computation time **(Martelet et al., 2002)**.

The method used in this work is based on modelling the area to be corrected using rectangular prisms and calculating the gravitational influence of each of these prisms at each observation point. The construction of these prisms, which approximate the topography, therefore requires the use of a precise digital terrain model (DTM) extended up to 167 km around the gravity stations. The prisms can be extended either to the height of the station or to the reference ellipsoid. In the first case, the effect of the prisms is added to the simple Bouguer correction, while in the second case it already includes the effect of the latter correction. It is this option, the terrain correction presented above, that will be used in the following. This choice is justified, firstly, by the desire to compare the results of this terrain correction method using prism modelling with those of the simple Bouguer correction. Secondly, the extension of the prisms to the ellipsoid means that the prisms do not have to be redefined for each station, which significantly reduces the computation time.

According to **(Nagy et al., 2000; Nagy et al., 2002; Heck and Seitz, 2007)**, the influence of a prism of known density ρ at $[x_p ; y_p ; z_p]$ with respect to an observer at $[x; y; z]$ is determined by eq. (25):

$$c = G \int_{-\infty}^{\infty} \int_{-\infty}^{\infty} \int_{z=H_p^0}^{z=H_{(x,y)}^0} rho\,(x,y,z) \cdot \frac{z - H_p^0}{\sqrt[3]{\left(x - x_p\right)^2 + \left(y - y_p\right)^2 + \left(z - z_p\right)^2}} dxdydz \tag{25}$$

where $H_p^o$ is the ellipsoidal height. So, given the geometry of the prisms and their distance from the observer, the Nagy potential is determined by solving equations shown in Fig. 14:

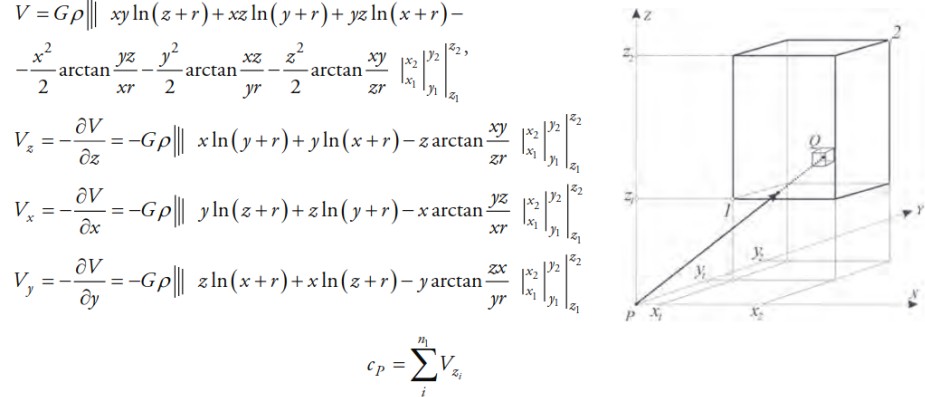

**Figure 14: Nagy potential equations, from (Nagy et al., 2000; Nagy et al., 2002).**

Finally, the terrain correction at an observation point is equal to the sum of the vertical contributions of the Nagy potential at that point. The rectangular prism method can also account for the effect of Earth curvature by applying a transformation matrix to prisms whose prism-to-observer distance is greater than 15 km **(Smith, 2000; Smith, 2002; Heck and Seitz,**



**2007; Makhloof, 2008**).

As mentioned above, the terrain correction is determined on the basis of a topographic grid extended to a reference distance of 167 km around the gravity stations. The accuracy aimed for by this correction obviously depends on the accuracy and sophistication of the DTM used. In order to use the most accurate DTM possible while optimising computational time and
memory, the terrain correction using prism modelling has been divided into two successive stages. The first is a terrain correction based on a 25 meter grid extended to 167 km, hereafter referred to as 'regional terrain correction'. A second terrain correction, called 'local terrain correction', based on a finer grid, detailed to the meter and extended to 6.65 km around the stations, was then applied. The general method based on the above expressions was implemented during these two stages using algorithms developed by the Python library (**Harmonica, 2023**).

Another advantage of using this numerical method is that it is possible to incorporate a numerical density model into the correction by assigning different densities to the different prisms. This is particularly interesting in the case of a sedimentary basin consisting of young, low-density lithologies trapped in a much denser surrounding rock, as in the case of the Mons Basin. However, this advanced treatment requires precise knowledge of the geometry of the basin floor and the densities to be assigned to the layers in the model. As there are still considerable uncertainties regarding the latter, it was decided to use
a simple model with a constant density. In principle, this should correspond to the densities of the different rocks above the reference ellipsoid, but due to the lack of knowledge of the subsurface, it is common to use a value of 2670 kg/m³ for all gravity stations. This value is generally used in gravimetry as it corresponds to the average density of surface rocks in a continental context (**Hayford and Bowie, 1912; Hinze, 2003**). Nevertheless, the model used takes into account the prisms located below the reference height, especially those related to bathymetry. In this specific case, the problem is solved by
assigning a density of -1.67 to these prisms, which is obtained by subtracting the bedrock density from that the water density, here approximated to be 1000 kg/m³ (**Vajda et al., 2008**).

**Regional terrain correction elevation grid generation**

As a result, a great deal of effort went into the construction of the two DTMs. The first DTM, measuring 25x25 m² and covering 167 km, was constructed using the following sources

- The 2013-2014 1x1 m² DTM of Wallonia, whose reference coordinate system is EPSG 31370 (Lambert 72) and whose reference height system is the Second General Levelling of the Kingdom (EPSG:5710). The owner of the data is the Service Public de Wallonie (SPW).
- The 2004 DTM of Flanders 25x25 m² "DHM-Vlaanderen, raster, 25 m", whose reference coordinate system is EPSG 31370 (Lambert 72) and whose reference height system is the Second General Levelling of the Kingdom
(EPSG:5710). Owner of the data: Agentschap voor Geografische Informatie Vlaanderen.
- European 1 arc second bathymetry, whose reference coordinate system is EPSG 4326 (WGS84) and whose reference height system is the European Vertical Reference Frame 2000 (EPSG:5730).
- The 25x25 m² "BD ALTI®" DTM of France, whose reference coordinate system is EPSG 2154 (Lambert 93) and whose reference height system is the Nivellement Général de la France - IGN69 (EPSG:5720). The owner of the
data is the Institut National de l'Information Géographique et Forestière (IGN).
- The Copernicus 1 arc second DTM whose reference coordinate system is EPSG 4326 (WGS84) and whose reference height system is the European Vertical Reference Frame 2000 (EPSG:5730).



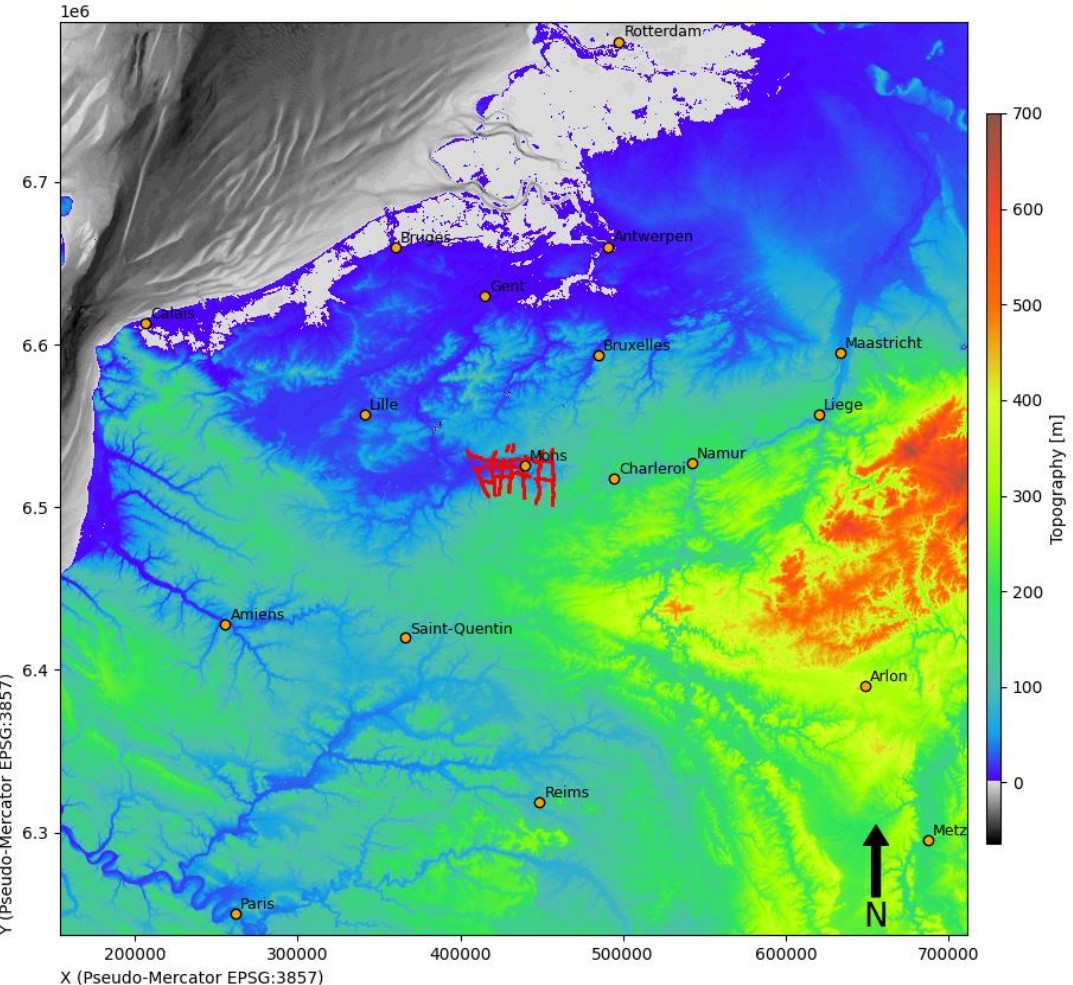

**Figure 15: DTM extended to 167 km around gravity stations with 25x25 m² resolution**

To create the unique height grid, each of these source data was first reprojected into a common geographic and height coordinate system, namely the Pseudo-Mercator projection (EPSG:3857) and the European Reference Frame for Heights EVRF2000 (EPSG:5730). The choice of the Pseudo-Mercator projection was justified by the need to have a reference projected coordinate system valid over the entire DTM area, which would not have been the case with a conical projection such as Lambert 72, valid over Belgium, or with a coordinate system such as WGS84.

Reprojections between vertical reference systems were performed using correction grids or the Python library Pyproj. EVRF2000 is the only European system for which transformations from different source reference systems have been possible using this library. Projections to more recent European reference frames (EVRF2008 and EVRF2019) are not currently implemented in this library. Each DTM was then aligned and resampled on a regular 25x25 m² grid and finally merged into the grid shown in Fig. 15. The order in which the DTMs were merged was the order in which they were
presented above. This choice was motivated by the proximity of the DTMs to the study area and their original resolution.

**Local terrain correction elevation grid generation**

The second DTM used to calculate prism effects corresponds to the rectangular convex envelope with a 6.65 km buffer around the gravimetric stations. The extent of this second DTM was limited to 6.65 km, corresponding to the first 10 zones of Hammer's method **(Hammer, 1939)**. In practice, the extent that optimises computational time in relation to the gain in





local terrain correction will be tested below. As the extent zone is located on both sides of the Franco-Belgian border, the 1x1 m² height grid was constructed on the basis of the following sources

- The Walloon 1x1 m² DTM of 2013-2014 (EPSG:31370+5710) with an altimetric accuracy of ±0.12 m, corresponding to a maximum error of about 37 µgal.
- The France 1x1 m² DTM "RGE ALTI®" (EPSG:2154+5720) with an accuracy of ±0.2 m, corresponding to a

maximum error of about 61.72 µgal.

Once again, particular attention was paid to the standardisation of the geographic and altimetric coordinate systems. For the former, EPSG 31370 was chosen to limit the deformations associated with the reprojection of the Walloon DTM on which most gravity stations are based. This time, the small size of this altimetric grid means that a conic projection such as Lambert 72 can be used, unlike the grid used for the regional terrain correction. However, in order to carry out the local

terrain correction and to compare the two grids, it was necessary to reproject the part of this first grid that was in the same spatial extent into the EPSG 31370 coordinate system. The altimetric coordinate system chosen is EVRF 2000, for the same reasons as above. The results of the of the cross-border DTM is shown in Fig. 16.

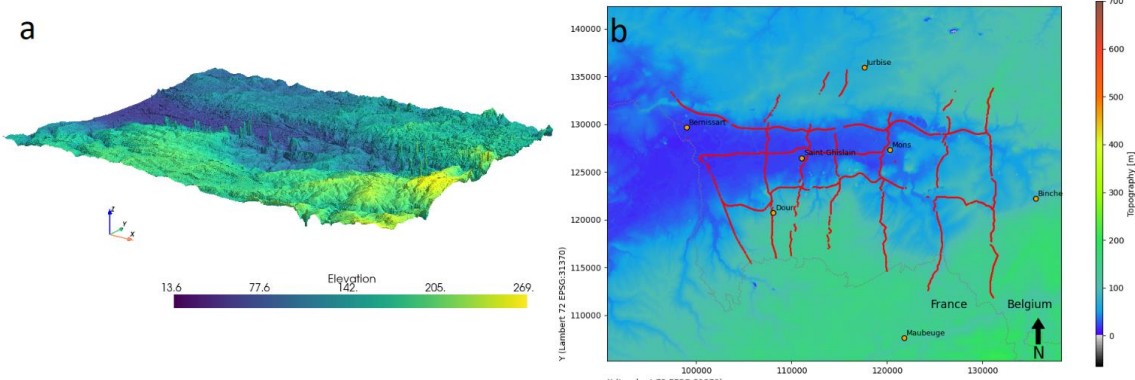

**Figure 16: 3D representation of the Franco-Belgian DTM used in the local terrain correction (a). Vertical exaggeration x10. DTM limited to 6.65 km around gravity stations, resolution 1x1 m² (b).**

**Application of regional and local terrain corrections ($C_{\text{Reg. terrain}}$ et $C_{\text{Loc. terrain}}$)**

Firstly, rectangular prisms were constructed on the basis of the first regional terrain correction grid. In particular, these prisms, which have 25 m on each side at the top and bottom, are modelled between the EVRF2000 reference ellipsoid and the 25 m resolution DTM. The density model presented was also integrated so that each prism was assigned a density of 2670 kg/m³ or -1670 kg/m³ depending on its location relative to the EVRF2000 reference plane.

Next, the prisms used in the local terrain correction were also constructed. These other prisms, with a base of 1 m², are modelled between the two DTM grids presented above, after reprojection of the first grid in CRS EPSG 31370. To ensure that the two methods are properly coupled, it is necessary to construct a suitable density model that assigns a density to each prism according to the relative position of its top and bottom surfaces.



As shown in Fig. 17, the regional terrain correction corrects the effect of terrain up to the reference height of the 25x25 m²
DTM. However, the real topography, which is best approximated by the 1x1 m² DTM, is either above or below the surface
of the first altimetric grid. In the first case (S1), the effect of these new prisms must be added to account for the mass deficit
not accounted for in the regional terrain correction, while in the second case (S2) it must be subtracted.

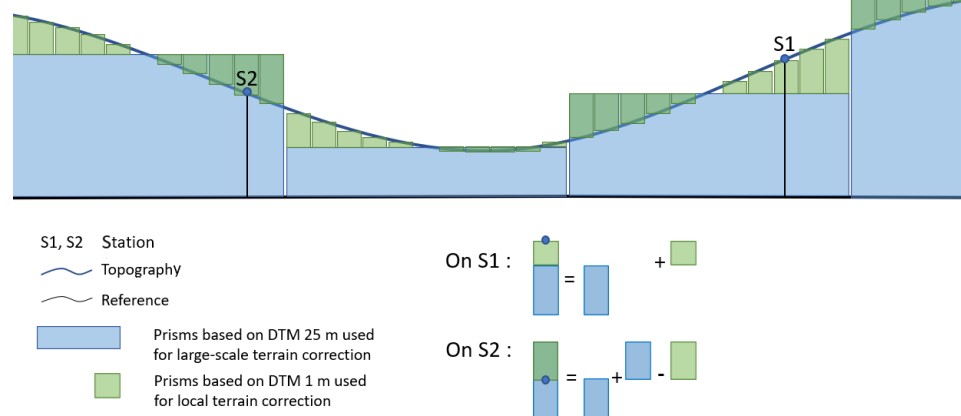

**Figure 17: Schematic representation of the coupling between local and regional terrain corrections.**

This feature is implemented when building the density model by assigning a positive density to prisms whose base
corresponds to the 25x25 m² DTM, and a negative density to those whose base is formed by the 1x1 m² DTM. A quick look
at Nagy's formula above shows that a negative density does indeed subtract the effect of a prism. Based on the above, the
density of these new prisms therefore varies between 4 possible values, $\pm 2.67$ and $\pm 1.67$, although the last two values were
not encountered in practice as the bathymetry was well beyond 6.65 km. It was also necessary to determine the height at
which the gravimetric influence of the prisms involved in these two corrections would be determined. In general, it is clear
that it is necessary to impose the same height on stations for regional and local terrain corrections. For this reason, the
height of the 25 m resolution DTM will never be used. Intuitively, the topographic survey of the stations carried out
simultaneously with the gravimetric campaign seemed a good choice. However, there are inevitable differences between the
surveyed heights and those of the 1m resolution DTM, as shown in Fig. 18.

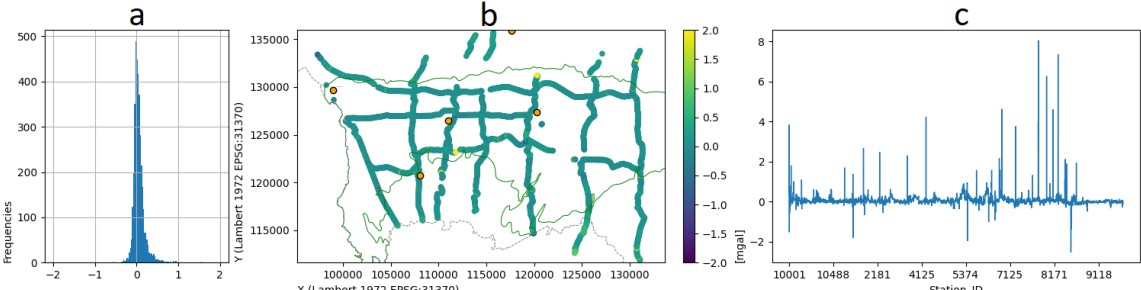

**Figure 18: Histogram (a), regional map of spatial distribution (b) and temporal distribution (c) of elevation differences between**
**GPS and the Local terrain correction elevation grid.**

The case of station 10001 was chosen to illustrate the impact of elevation selection in modelling. The elevation of this
station in 2022 is 33 m. However, the 1 m resolution DTM indicates an elevation of 32.52 m, while the 25 m resolution
DTM indicates an elevation of 33.53 m. The local terrain correction was performed in a 2 km zone around this station,
considering different elevations for this station, and the result is shown in Fig. 19. It is now clear that the choice of station
height reference significantly alters the result.





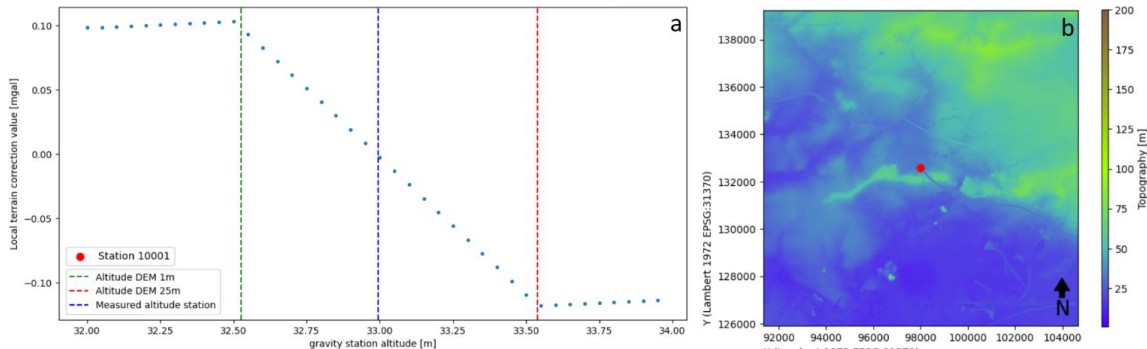

**Figure 19: Influence of the elevation taken into account in the calculation of the local terrain correction at station 10001 and the transfer of the measured elevation (in blue) and the elevations of the first (in red) and second (in green) elevation grids (a). Location of gravimetric station 10001 (in Bernissart along the canal) on the DTM at 1m resolution (b).**

In this particular case, using the measured elevation gives a result close to zero, a consequence linked to the central position of the measured elevation within the prism formed by the two DTMs at the position of station 10001. On the basis of these observations alone, it would be possible to decide which elevation to use in the two terrain corrections, that of the 1m resolution DTM. However, apart from these observations, the very principle of this correction justifies this choice, since it is designed to correct, in relative terms, the topography surrounding a station. This topography is more accurately modelled by the 1m resolution grid, so it's clear that using this height is more appropriate. What's more, this surface models the

topography with a perfectly acceptable altimetric accuracy of 0.2 m (the maximum inaccuracy among the source DTMs).

With regard to grid size, the reason for extending the regional terrain correction grid to 167 km was mentioned. However, the extent of the local terrain correction grid could possibly be reduced to reduce computation time. Once again, station 10001 was selected to model the effect of the local terrain correction grid extent. Figure 20 shows that the accuracy targeted by the correction is quickly achieved with a grid extended to a few hundred metres. In particular, for a grid extended to 600

m around the station, the difference between the result obtained and that of the 6.65 km grid is less than 0.1 µgal, the accuracy of Scintrex CG6. To be on the safe side, an extension of 2 km around each station was chosen.

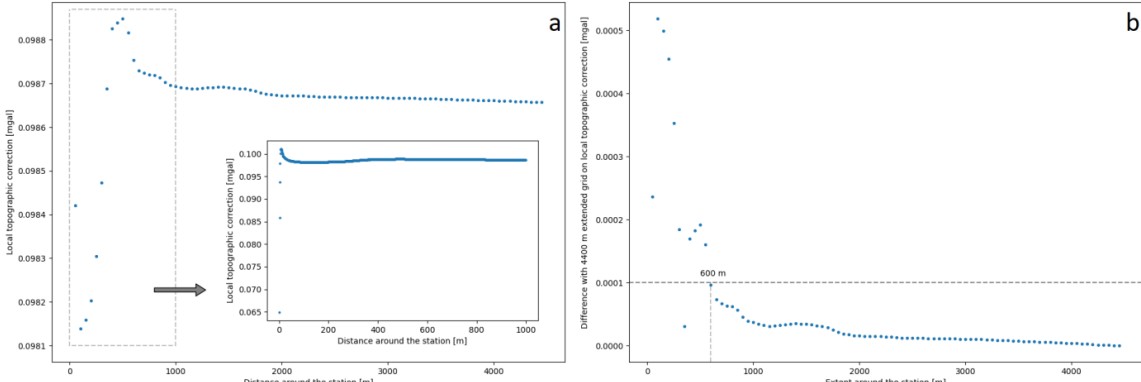

**Figure 20: Result of the local terrain correction as a function of the extent of the grid used in the prism modelling for station 10001 (a). Difference between the result of the local terrain correction obtained with a grid extended to 6.65 km and the extension intended for station 10001 (b).**

Finally, regional and local terrain corrections were applied. The results shown on the maps below in Fig. 21 represent the regional and local terrain effects respectively. In the Fig.21.a, the regional terrain effect is quite significant, with a wide range from 1.62 to 17.04 mgal. This is mainly due to the the simple Bouguer effect, whose range of values is quite similar. Indeed, the simple Bouguer and Bouguer spherical cap corrections were not applied for MoreGeo since it was decided to integrate them into the terrain correction but, the ranges of these two corrections have been evaluated with a constant

correction density of 2.67. These are between -1.67 mgal and -17.35 mgal and between -1.69 mgal and -17.56 mgal,



respectively.

In fact, the regional terrain effect is greatest on the plateau areas where the altitude is highest and the relief is less rugged (for example, north and south of profile 7), resulting in a high simple Bouguer effect and a relatively low topographic effect. In the shallow valley of the Haine river in the centre of the map, where the altitude is low, such as north of Saint-Ghislain, the terrain effect is minimal as the topographic and simple Bouguer effects are also greatly reduced.

On the Figure 21.b, the local terrain effect has minor importance, limited to a range between -0.05 and 0.66 mgal. Its influence is all the greater in areas where the relief is close and strongly marked, such as in railway trenches (e.g. to the west of profile 10), in areas close to spoil heaps, bridges and haul roads (encountered by lines 9, 10 and 8). It is also worth noting that the choice of station 10001 to assess the impact of the grid extent involved in the local terrain correction was a wise one, as it is located in the area where the maximum effect was found.

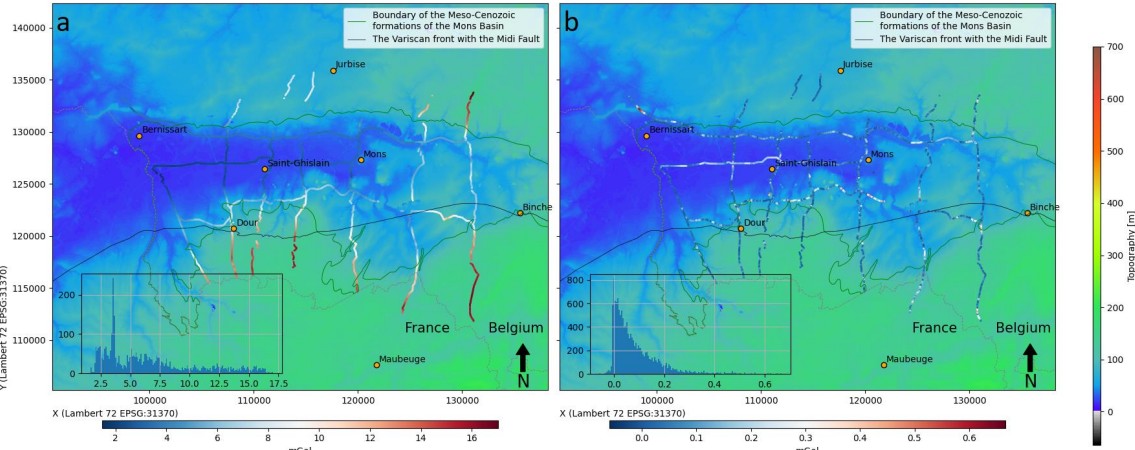

**Figure 21: Regional terrain effect map calculated with a reduction density of 2.67 and the 25 m resolution elevation grid extended to 167 km around the gravimetric stations (a). Local terrain effect map calculated with a reduction density of 2.67 and the 25 and 1 m resolution elevation grids extended to 2 km around the gravimetric stations (b).**

By adding the local and regional terrain corrections and subtracting the spherical Bouguer effect determined above, it is possible to obtain the real topographic effect. As estimated by previous studies carried out in the region **(Battaille, 1967; Leclerc, 1960)**, it is quite small, although not negligible, averaging -0.14 mgal and reaching up to 0.4 mgal. Figure 22 shows the gravity anomaly obtained after applying these two terrain corrections and the spherical Bouguer correction along a profile. The anomaly obtained after applying the spherical Bouguer correction and that obtained after applying the regional terrain correction are shown in purple and blue respectively. Green represents the anomaly after application of the local and regional terrain corrections. The differences between the green and the purple curves are therefore due to the topographic effect.

This profile (No. 7, running S-N from Jeumont in France to Le Rœulx in Belgium), which was mainly acquired along a Ravel, also illustrates the effect of local terrain correction. In fact, two areas show significant differences between the blue and green curves. This demonstrates the inability of the 25x25m DTM to capture all topographic features, such as bridge crossings, and the importance of correcting the topography with a more accurate DTM. Finally, in addition to the importance of this terrain correction, it allows us to eliminate "pseudo-anomalies" that are actually related to the topography, which is very evident on profile no. 7.





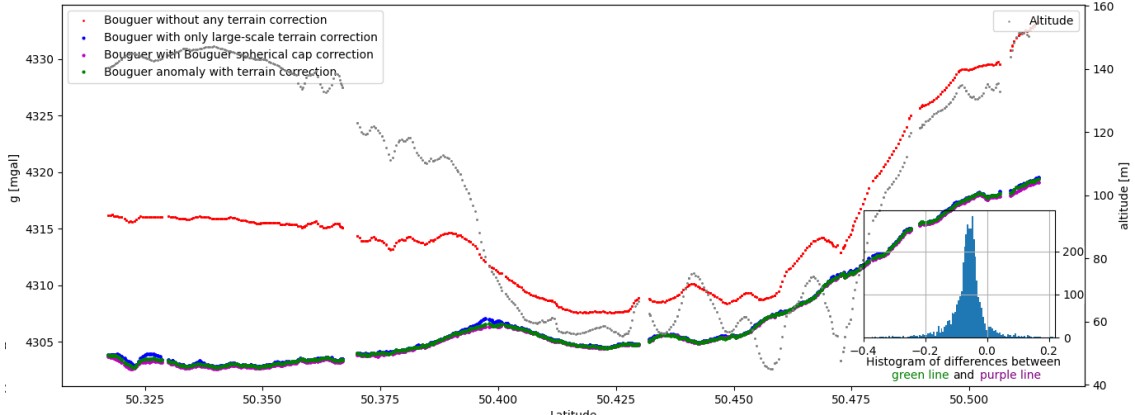

**Figure 22: Comparison between terrain correction (green), regional terrain correction (blue) and spherical Bouguer correction (purple) applied to gravity measurements on profile S-N n°7 from Jeumont to Le Rœulx. Also shown are the gravity measurements before terrain correction (red) and the elevation profile (grey) based on the 1x1 m² elevation grid.**

**Latitude correction ($C_{Lat.}$)**

The Earth's rotation exerts a centrifugal force on an object at its surface, the intensity of which varies with latitude. This effect inevitably affects gravimetric measurements. Therefore, as latitude decreases and we move southwards, the centrifugal effect increases, which reduces the intensity of the gravimetric measurements. The formula used in this study to correct for this effect is that of **(Moritz, 1980)** :

$$C_{\text{Lat}} = A \cdot \frac{1 + B \sin^2(\varphi)}{\sqrt{1 - C \sin^2(\varphi)}} \tag{26}$$

Where φ is the latitude of the station and ge, k and e2 are constants (A = 978032.67714, B = 0.00193185138639, C = 0.00669437999013).

This correction also takes into account the ellipsoidal shape of the Earth, and in particular the radius of the Earth, which varies with the position of the observer. The effect is therefore particularly visible for N-S lines. Figure 23 illustrates the effect of this correction on the gravimetric measurements made along the line n°2. In this example, this correction is applied with respect to a base station, which is why the sign varies according to the position of the measurement station with respect to the base station. The value of this correction can therefore be up to a few mgals.

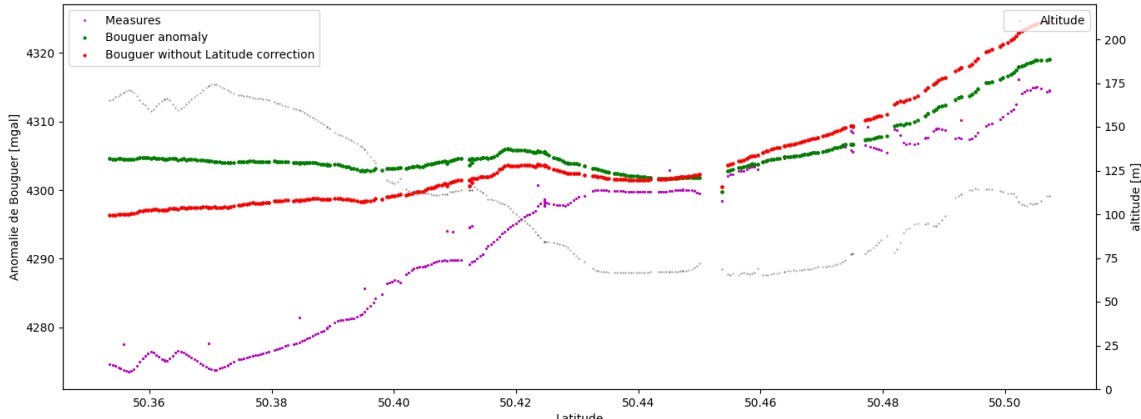

**Figure 23: Effect of latitude correction on processed gravity data line 2, S - N. The topographic survey (black), the raw**



measurements (purple), the Bouguer anomaly with (green) and without latitude correction (red) are shown.

**Atmospheric correction (C<sub>Atm.</sub>)**

The measurements made by the gravimeter do not take into account the atmospheric mass above the measuring station. The atmospheric correction consists in correcting the effect of this mass according to the formula of **(Ecker and Mittermayer, 1969)**.

$$C_{Atm} = 0.874 - 9.9 * 10 - 5h + 3.56 * 10 - 9h^2 \qquad (27)$$

Where h is the height in metres relative to the GRS80 ellipsoid in the WGS84 geodetic datum. The effect of this correction is quite small, not more than a dozen µgals over the whole campaign.

**Temperature and instrument tilt corrections (C<sub>Temp.</sub> et C<sub>Tilt</sub>)**

The Scintrex CG-6 Autograv™ is a gravimeter that incorporates a number of automatically calculated corrections. It takes into account four corrections: temperature, instrument tilt, tides and spring fatigue (by evaluating the instrument drift). In the processing scheme implemented for this campaign, only the first two corrections were retained because they cannot be easily calculated a posteriori unlike the last two corrections. In fact, the drift correction evaluated by the instrument over the
three years separating the two campaigns turned out to be too large and totally erroneous.

**Gravity anomaly from MoreGeo 2019-2022 survey**

By applying all the gravity corrections presented above to the instrumental measurements, it is now possible to extract the gravity anomaly. Thus, for the 2019-2022 MoreGeo campaign, the anomaly is calculated as follows:

$$\text{Gravity anomaly} = g_{Measured} + C_{Tide} + C_{Drift\ intra-period} + C_{Drift\ inter-periods} + C_{Free\ air} + C_{Reg.\ terrain} +$$
$$C_{Loc.\ terrain} + C_{Lat.} + C_{Atm.} + C_{Tilt} + C_{Temp.} \qquad \mathbf{(28)}$$


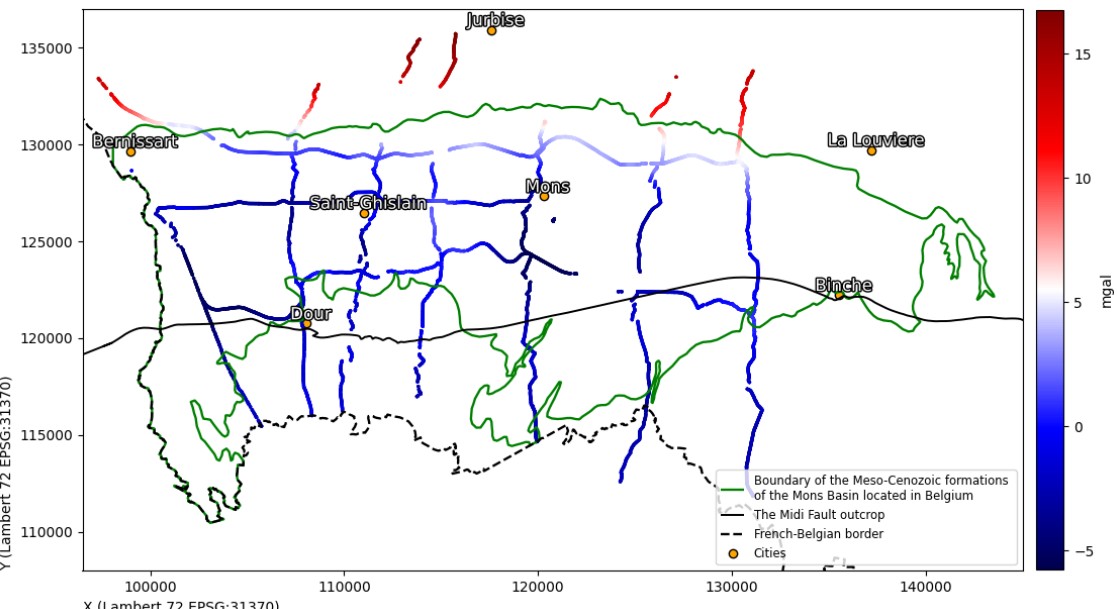

**Figure 24: Map of the gravity anomaly measured by the stations of the MoreGeo 2019-2022 campaign.**

Figure 24 shows the result of applying these corrections to MoreGeo's gravity stations. All temporal, topographic, altimetric and latitudinal effects have now been considered to reveal the gravity anomaly associated solely with variations in the
density of the subsurface masses. At first sight, the gravity anomaly follows a general north-south trend, as suggested by the



maps in **(Everaerts and De Vos, 2012; Dupont, 2021)**. However, lines 1 to 7 show minima of the gravity anomaly that are more aligned with the Mezo-Cenozoic formations of the Mons Basin, which are outlined in green in this figure. The Variscan front and its major accident, the Midi fault, are also shown.

At this point it is important to discuss the accuracy of the MoreGeo gravity anomaly assessment. Obviously, in order to assess this accuracy, it is necessary to take into account all sources of uncertainty, whether related to the measurement campaign or to the processing steps. Thus, in the context of MoreGeo and the gravimetric processing presented in this work, the root mean square error, which corresponds to the inaccuracy obtained on the gravimetric anomaly $e_B$, is calculated with eq. (29):

$$e_B^2 = e_{Survey}^2 + e_{GPS}^2 + e_{Drift}^2 + e_{Topo}^2 \qquad (29)$$

$e_{Survey}$ is the imprecision associated with the measurement of the anomaly. As a reminder, the precision aimed for by the Scintrex CG-6 was discussed in the chapter dedicated to the acquisition campaign and is of the order of 5 µgal. In practice, the value of 10 µGal was chosen because it corresponds to the repeatability of measurements obtained by operators on the field. $e_{GPS}$ is the inaccuracy associated with GPS geographical positioning measurements. It is of the order of 5 mm maximum (RTK fixed), which gives an insignificant uncertainty of ±4 ngal on the latitude correction (the latter varies on
average by 0.8 µgal/m at the average latitude of the study area).

The elevation error of the $e_{Topo}$ stations used in the terrain and elevation corrections is evaluated on the basis of the accuracy of the 2013-2014 Walloon 1x1 m DTM, which is 12 cm, corresponding to a maximum error of about ±37 µgal. Finally, $e_{Drift}$ is the uncertainty of the drift correction, evaluated at 10 µgal on the basis of the dispersion of the measurements at the three campaign bases calculated after drift correction. This gives

$$e_B^2 = 10^2 + 0.004^2 + 10^2 + 37^2 \ \mu gal^2 \qquad (30)$$

$$e_B = 39.61 \ \mu gal \qquad (31)$$

The root-mean-square error evaluated for this MoreGeo campaign is therefore 40 µgal, well below the initial target of 100 µgal.

**Integrate MoreGeo with legacy databases**

The current spatial distribution of MoreGeo stations along the profiles allows us to obtain the anomaly along these lines with a high degree of accuracy. However, this spatial distribution is oversampled along the direction of the lines, which inevitably results in a loss of anomaly sensitivity outside this direction. This feature severely limits the ability to interpolate the anomaly outside of the lines and therefore any attempt at mapping. In order to improve the regional imaging of the gravity anomaly, it is therefore necessary to first integrate all gravity data available in the study area.

Furthermore, as discussed in the introductory section, the refinement of the regional gravity anomaly mapping requires the collection and standardisation of all gravity data available in the area. To this end, the relevant gravity campaigns carried out since the middle of the 20th century have been surveyed and three gravity databases containing these older campaigns have been integrated with the MoreGeo 2019-2022 campaign data.

In particular, these include the Belgian gravity database published by the Royal Observatory of Belgium (ROB) **(Verbeurgt**
**et al., 2019)**, data from the 1967 Battaille campaign carried out by Battaille Frères society for CGG **(Battaille, 1967)**, and the French gravity database **(BRGM, 2023)**. The map shown in Fig. 25 represents these different data sources within a 20 km radius of the MoreGeo campaign data. This area was selected as the final study zone used to interpolate the gravity anomaly.



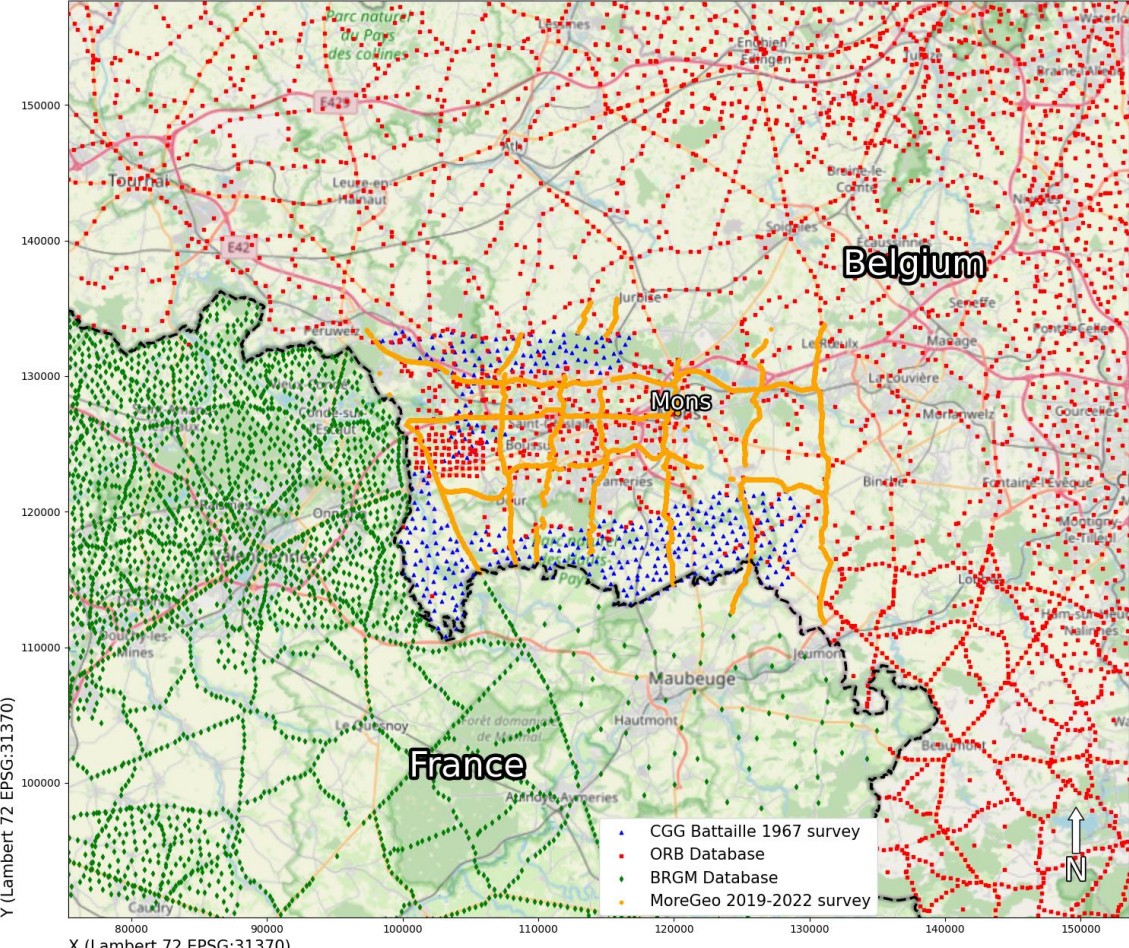

**Figure 25: The different sources of gravity data integrated into the Moregeo campaign data in this work. In red: the Belgian gravimetric database, published by the Royal Observatory of Belgium, from (Verbeurgt et al., 2019). In green: the French gravimetric database from (BRGM, 2023). In blue: The gravimetric campaign carried out by Battaille Frères society in 1967, from (Battaille, 1967). ©OpenStreetMap Distributed under the Open Data Commons Open Database License (ODbL) v1.0.**

However, there are four main reasons why all these different databases cannot be combined into a single regional database, as shown in Fig. 26. Firstly, all the corrections developed above and applied to the MoreGeo campaign have not necessarily been applied to these three older datasets. For example, the ROB and BRGM databases have only incorporated the temporal effects of tides and drift into their gravity data. The second is related to corrections that have already been applied to these datasets and which require a step back. In particular, the full Bouguer correction, for which the reduction density introduced varies depending on the data source. The modular nature of the many methods and corrections implemented in this work will allow us to address these first two points.

The third reason is related to the geographical and altimetric reference points to which the data sources are referenced. For example, the BRGM data are referenced to French benchmarks, while other data sources are referenced to Belgian benchmarks. Finally, a problem with the georeferencing of the gravimetric stations of certain campaigns present in the ROB database was highlighted during this work.



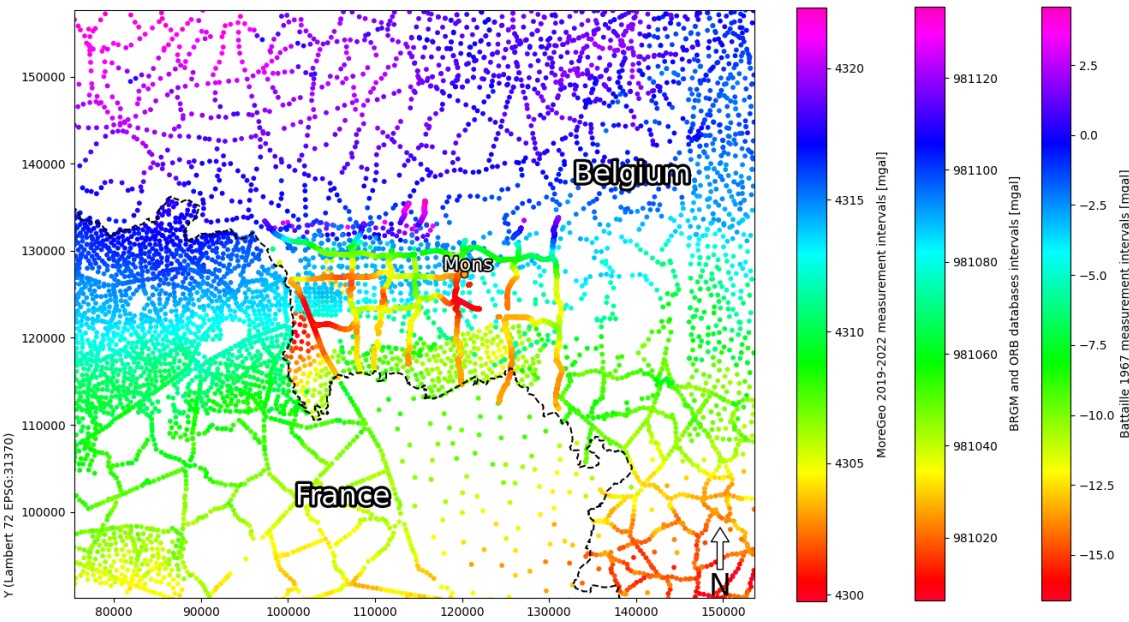

**Figure 26: Map of the raw collection of all gravity data retrieved from the ROB and BRGM databases, as well as from the MoreGeo 2019-2022 and Battaille 1967 campaigns, used in the regional gravity anomaly mapping.**

**The gravity database of Belgium**

The first database to be integrated is the Belgian gravimetric database, published by the Royal Observatory of Belgium. **(Verbeurgt et al., 2019)** present the work carried out in preparation for the publication of this database, which involved the collection and standardisation of gravimetric data obtained over the Belgian territory during numerous campaigns (±69,000 stations). In the study area, the following campaigns were selected *2nd Belgian gravimetric network* (1948), Mons (1960), *Dinant basin* (1962), extension 1948 (1972), *Geraardsbergen* (1993), *Central Belgium* (1996) and Philippeville (2000). All data in this database are topographically referenced, i.e. only the temporal corrections for lunisolar tides and drift have been applied.

However, before applying the other gravimetric corrections, it was necessary to correct the position of the stations belonging to certain campaigns. In particular, for the Mons (1960) and Extension 1948 (1972) campaigns, the two closest to MoreGeo, it was necessary to take the original documents from **(Leclerc, 1960)** and **(Battaille, 1968-69)**, georeference them and check the positions of the gravity stations from these two campaigns. As a result, the positions of the stations from these two campaigns were corrected, sometimes by as much as 150 m.

Latitude, atmospheric, altitude and terrain corrections were then applied to the data. This was done using the original gravity station heights recorded at the time of acquisition, as these were already available in the database. However, these heights were first reprojected from their original altimetric system (EPSG:5710) to EVRF2000 (EPSG:5730) in order to standardise the altimetric reference with the MoreGeo campaign data. The result of these corrections is shown in Fig. 27 and represents, along a south-north section of the data before (in red) and after (in green) the application of the corrections.



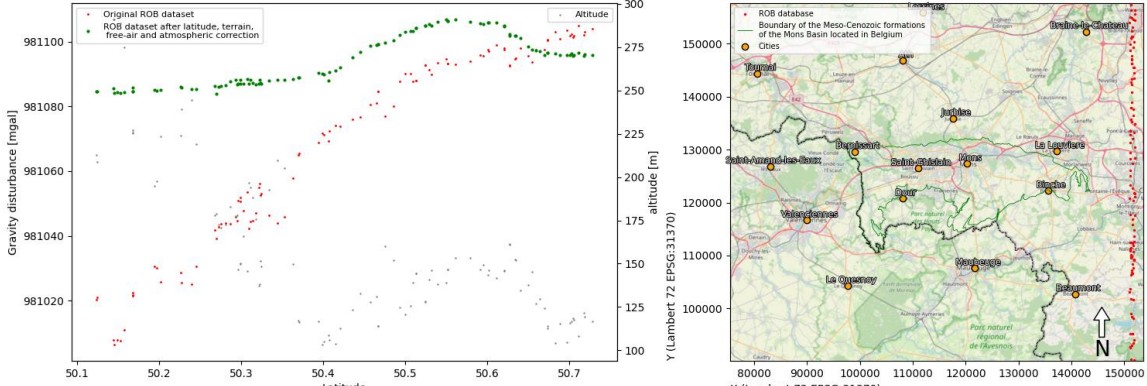

**Figure 27: Data from the ROB database before (red) and after (green) application of latitude, atmospheric, altitude and terrain corrections along a south-north section between Froidchapelle and Braine-l'Alleud. ©OpenStreetMap Distributed under the Open Data Commons Open Database License (ODbL) v1.0.**

To complete the integration of these data with those from MoreGeo, one of the two datasets had to be realigned with respect to the other. The ROB database originally provided absolute measurements of gravity acceleration relative to the topography. It was therefore decided to align the measurements from the MoreGeo campaign with those from the ROB. This was done by identifying the two closest pairs of MoreGeo and ROB stations, 5.3 m and 6.8 m apart. As the two MoreGeo stations were acquired three times, these two pairs of stations form a set of 6 offsets, the average of which was 745 used to recalibrate MoreGeo to the ROB data.

**The Battaille 1967 survey**

The Battaille 1967 dataset is rather unusual in that, although it was acquired by CGG for Battaille Frères in the same way as the Battaille 1968-69 campaign, this 1967 dataset is not included in the ROB database, unlike its 1968-69 counterpart. In this sense, the integration of this 1967 Battaille campaign into the regional mapping of the gravity anomaly is also an 750 original aspect of this work.

However, the documents relating to this **(Battaille, 1967)** campaign are scarce. Indeed, no documents relating to this campaign seem to have been preserved in the CGG archives **(Dupont, 2021)** and only a campaign report and the final map of the anomaly calculated at the time with the reduction density used were found in the archives of the Belgian Geological Survey. The gravity anomaly obtained at that time was calculated with a simple Bouguer correction using a reduction 755 density of 2.1. To standardise the corrections, it is therefore necessary to subtract this correction and to apply a terrain correction with a reduction density of 2.67.

To do this, the exact height of the stations is required. To this end, the 1x1 m² elevation grid constructed in the previous section for the local terrain correction was used to recover the precise elevation of the stations. However, this implies that the altitude of these stations and the surrounding topography has not changed in 57 years. However, it is well known that the 760 Borinage region experienced significant subsidence during the 20th century due to mining subsidence **(Ghiste and Albert, 1980; Leroy, 2014)**.

IIt was therefore decided to exclude stations located within the influence zone of these mines, where subsidence was likely to have occurred. To determine this zone of influence, the extent of the mining zones mapped in **(Delmer, 1977)** was used. A buffer zone of 1 km was selected around these claims to represent the area of land likely to have been affected by mining 765 subsidence. This extent was determined by taking into account the maximum depth of the workings around this zone (1000 m) and an angle of influence of 45°, which corresponds to the pessimistic situation regarding the influence of mining subsidence **(Piguet, 1997)**.



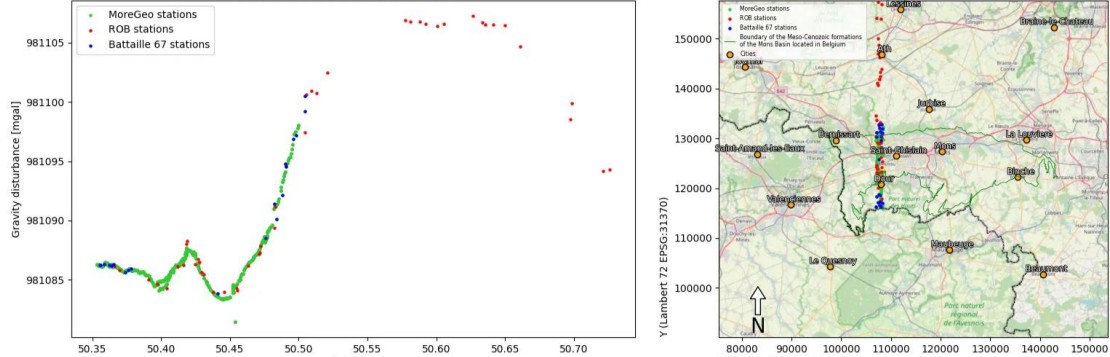

**Figure 28: Data from the ROB database (in red) from MoreGeo (in green) and the Battaille1967 campaign (in blue) after integration of the three datasets along a south-north section between Fayt-le-Franc and Lessines. Base map © OpenStreetMap**
**contributors. ©OpenStreetMap Distributed under the Open Data Commons Open Database License (ODbL) v1.0.**

On the basis of all these considerations, the effect of a simple Bouguer with a reduction density of 2.1 was replaced by a terrain correction with a reduction density of 2.67, as in the MoreGeo dataset. This old dataset was recalibrated to the two previous datasets in the same way as the MoreGeo-ROB dataset, based on the two closest station pairs between the datasets, this time at 10.6m and 10.7m respectively. The result of this integration is shown along a south-north section on the Fig. 28.
The latter shows in particular the accuracy of the integration of the first three datasets, as all gravity anomaly measurements between the datasets agree remarkably well.

**The gravimetric database of France**

Finally, there is the French gravimetric database **(BRGM, 2023)** to be integrated with the Belgian data. This database, published by BRGM, is regularly updated with data from new ground and airborne campaigns **(Martelet and Deparis,**
**2010)**. The integration of this dataset is quite similar to that of the ROB in that the gravimetric data are also said to be 'topographically referenced' and therefore only the latitude, atmospheric, altitude and terrain corrections with a density of 2.67 need to be applied to the data. However, this time the data is referenced to the French topography, whose reference height system is the Nivellement General de la France - IGN69 (EPSG:5720). It is therefore necessary to first reproject the stations to the EPSG:5730 of EVRF2000. It should also be noted that the terrain correction values applied to this dataset are
those published with this dataset in the publication **(Martelet, 2001)**.

Once again, these data were calibrated against the previous ones by determining the offset to be applied on the basis of the two closest pairs of stations between the data sets. This time, the distances separating these two pairs of stations are greater, namely 58 m and 59 m. The overall accuracy of this integration stage, hereafter referred to as $e_I$, is therefore limited by this final calibration. This has been estimated at 80 µgal on the basis of the maximum difference encountered between the six
offsets obtained by these two pairs of stations. This also allows us to evaluate the uncertainty of the MoreGeo gravity data integrated into this large database at 89 µgal, noted $e_i$ in the eq. (32).

$$e_i^2 = e_B^2 + e_I^2 = 39.61^2 + 80^2 \text{ µgal}^2 \qquad (32)$$

$$e_i = 89.26 \text{ µgal} \qquad (33)$$

The result of this integration is shown in Fig. 29. This last map represents the result of this integration stage, bringing
together 10140 gravity stations over an area of 5000 km².



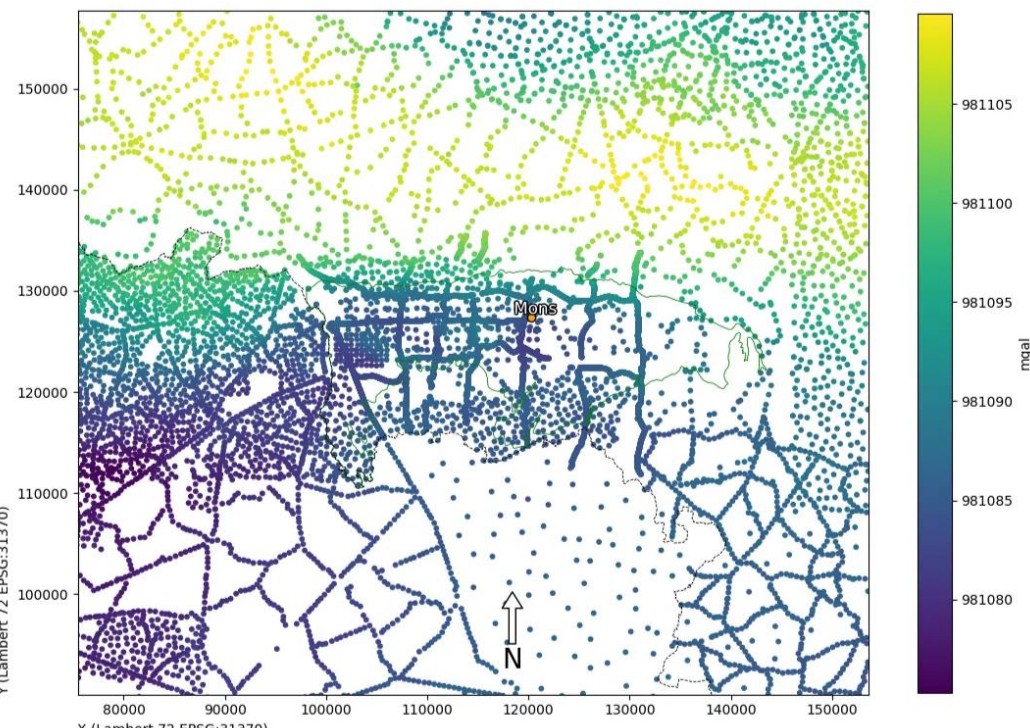

**Figure 29: Map showing data from the MoreGeo 2019-2022 and Battaille 1967 campaigns, and from the ROB and BRGM databases, after data calibration and standardisation.**

**New detailed regional map of the gravity anomaly**

**Design using the equivalent source method**

So far, the integrated data presented at the end of the previous section have been located on the 10140 gravity stations. This representation limits the mapping of the gravity anomaly and its processing, such as the calculation of gravity isovalue lines and gradient mapping. It is therefore essential to interpolate these data over this 5000 km² area.

Classically, interpolation methods such as inverse distance weighting or kriging methods **(Dupont, 2021)** are used to interpolate gravity data onto a regular grid. However, in gravimetry, the three-dimensional nature of the observed
phenomenon should be considered when mapping the anomaly. For example, differences in elevation between stations and the geometry of objects studied at depth are not taken into account in these classical interpolation methods. For this reason, the equivalent source method has been used in this work, as it allows these different three-dimensional aspects to be taken into account **(Dampne, 1969; Soler and Uieda, 2021; Harmonica, 2023)**.

This equivalent source method uses a process of least squares adjustment of point mass sources located in the vicinity of
each gravimetric station. The depth of these sources is fixed, while the density assigned to each is adjusted to reduce, in a least squares sense, the difference between the observed anomaly and the anomaly simulated by the set of equivalent sources. In addition to the depth of the sources, this method uses a second adjustable parameter, the damping parameter. This parameter controls the extent to which the simulated data are adjusted in relation to the observed data. This adjustment process therefore allows the noise to be smoothed by taking into account the uncertainty in the data.

In order to limit the number of sources involved and to reduce the computational time, the block-averaged equivalent source method introduced in **(Soler and Uieda, 2021)** and implemented in the **(Harmonica, 2023)** library has been used. For this purpose, it is necessary to define the size of a mesh for which a single equivalent source is modelled in the centre of the meshes containing at least one gravimetric station. This block mesh also makes it possible to reduce the spatial aliasing



caused by oversampling in a particular direction, especially that of the MoreGeo profiles. For this reason, a mesh size of 200 m was chosen, thus defining the resolution of the interpolation grid.

To determine the optimum damping/source depth combination, 28 combinations of these two parameters were tested (damping → [0.1, 0.5, 1, 5], source depth → [1000, 1250, 1500, 1750, 2000, 2250, 2500]). The selection criterion was based on the evaluation of $\chi^2$ calculated with gravimetric data from the MoreGeo campaign, as this was the only campaign where the measurement uncertainty $e_i$ evaluated in this paper was available. This criterion evaluates the average of the

squares of the uncertainty weighted differences between the observed anomaly and the anomaly simulated by equivalent sources. This statistical criterion is therefore preferable to the RMS, as it allows to take into account the uncertainty associated with the measurements, evaluated at 0.089 mgal for the MoreGeo campaign after its integration with the other databases.

$$\chi^2 = \frac{1}{N}\sum_i^N \left(\frac{d_i - f_i(m)}{e_i}\right)^2 = \frac{\Phi_d}{N} \tag{34}$$

Figure 30.a shows the result of this adjustment process using the equivalent sources method averaged per 200 m block, with the differences between the observed anomaly and the simulated anomaly at the gravimetric stations. This result was obtained by placing the sources at a depth of 1,250 m with a rather low damping parameter of 0.5, giving slightly more weight to data adjustment than to solution smoothing.

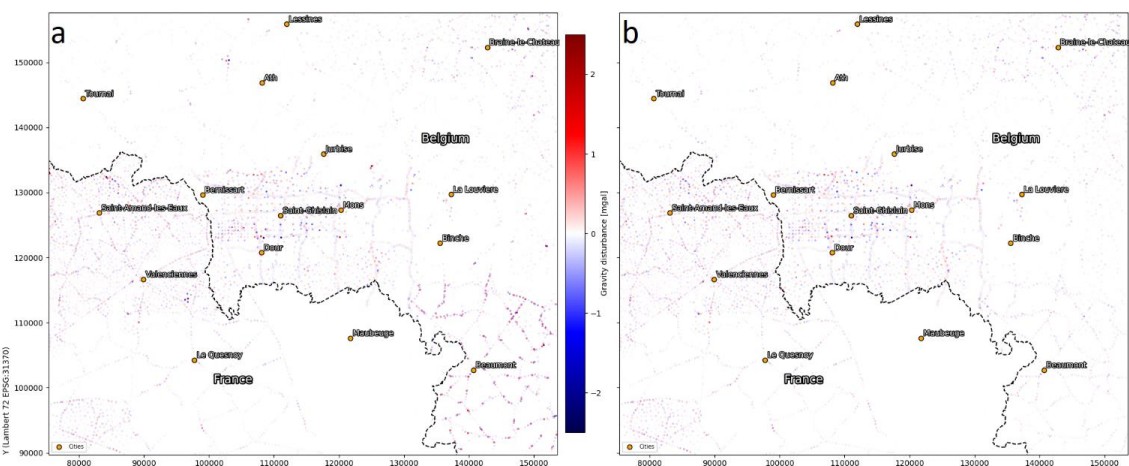

**Figure 30. Comparison between the observed Bouguer anomaly and the Bouguer anomaly simulated by the equivalent sources'**
**method averaged per 200 m block over the 10140 measurement stations (source depth: 1250 m; damping: 0.5) (a). Comparison between the observed Bouguer anomaly and the Bouguer anomaly simulated by the equivalent sources' method averaged per 200 m block after recalibrating the Dinant Basin campaign (1962) and eliminating outlier stations (source depth: 1250 m; damping: 1) (b).**

The $\chi^2$ and R² of this fit were estimated to be 1.138 and 0.9995 respectively, demonstrating its reliability in the least squares
sense, but also considering the inherent uncertainty of the data. However, Fig. 30.a highlights stations that deviate significantly from the chosen solution. For example, measurements from the Philippeville campaign (2020), located in the south-east of the map and belonging to the Belgian gravimetric database, systematically show significant positive differences, estimated at 1.35 mgal. This is related to a systematic difference between this campaign and the Dinant Basin (1962) campaign, acquired nearby along profiles, and therefore probably to a poor integration of one of these two
campaigns in the **(Verbeurgt et al., 2019)** database**.** In fact, these 1.35 mgal errors also appear in the central-eastern zone where the *Dinant Basin (1962)* and *Central Belgium (1996)* campaigns overlap, indicating that the *Dinant Basin (1962)* campaign would have been poorly calibrated with respect to the others.

Also the Extension 1948 (1972) and Mons (1960) campaigns **(Leclerc, 1960)** show higher than average positive and negative differences, probably related to the fact that these campaigns were less precise than the others, being older.



Another observation is that the MoreGeo campaign shows very small or even almost no differences, proving that direct modelling by equivalent sources can explain the measurements obtained by this high-precision campaign. These observations correlate with the mean and standard deviation of the differences between the observed and predicted anomaly for the MoreGeo data, which are 1.2 µgal and 96 µgal respectively.

Based on these observations, it was decided to proceed with a new phase of adjustment of the equivalent source model, after recalibrating the Dinant Basin campaign (1962) and eliminating outlier stations. This campaign was recalibrated by precisely determining the offset to be applied to this campaign, based on 15 pairs of stations separated by no more than 100 m. All measurements taken at stations from the 1948 (1972) extension campaign and those showing differences between the observed and simulated Bouguer anomaly greater than 2 mgal were considered outliers to be eliminated. In this way the optimal combination of attenuation and source depth was again determined using the $\chi^2$ of the MoreGeo stations. The result

of this second fitting is shown in Fig. 30.b. The and $R^2$ of this fit were estimated to be 1.031 and 0.998 respectively. The mean and standard deviation of the differences between the observed and predicted anomaly for the MoreGeo data remain virtually unchanged.

After the adjustment phase, a prediction step is performed to interpolate the gravimetric anomaly on the regular grid. The result is shown in Fig. 31. This figure therefore represents a new detailed mapping of the regional Bouguer anomaly, one of

the main results of this work. The gravimetric isovalue curves are plotted in 1.5 mgal steps to account for the wide extent of the anomaly in the region. The extent of the Meso-Cenozoic terrains of the Mons Basin recognised in Belgium and the line of origin of the Midi Fault are shown in green and black respectively.

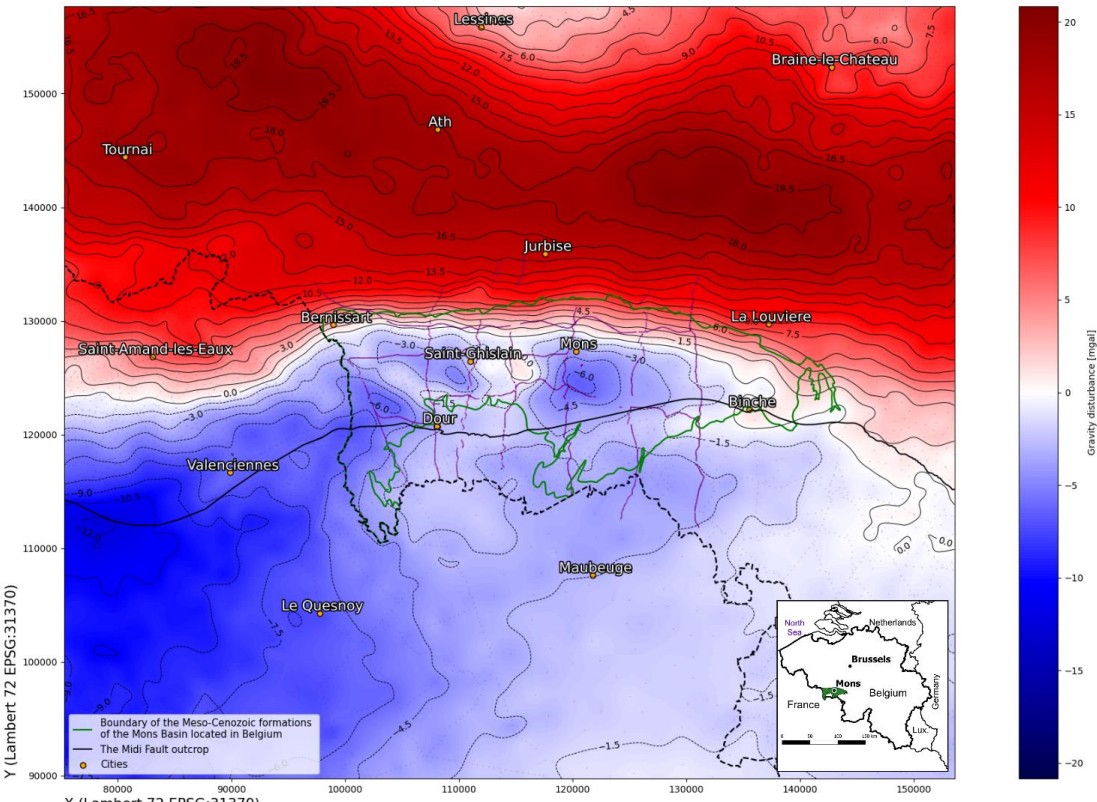

**Figure 31: New detailed 200 m grid gravity anomaly map and gravity isovalue curves for the French-Belgian Hainaut region.**

**Discussion and interpretation of the new gravity anomaly map**

At first sight, the general appearance of this new map follows broadly the same trend as the maps of **(Everaerts and De Vos, 2012)** and **(Minguely, 2007).** It would therefore be interesting to compare this new map with a gravimetric map



recompiled solely from the data used by these two references, i.e. the BRGM and ROB databases. This comparison would make it possible to assess the impact of the three areas of improvement studied in this work to resolve the gravity anomaly:

- Full control of the gravimetric processing chain, including terrain corrections - a first for Belgian data;
- The implementation and integration of the new MoreGeo 2019-2022 gravimetric acquisition campaign, characterised by a high spatial sampling rate and a low root mean square error of 39.61 µgal;
- Integration of the old data from the Battaille 1967 campaign.

These three areas of improvement have enabled us to produce a map with 25 times the resolution of the previous one and with a much greater number of measurements, this time covering the entire study area. In order to quantify these three
aspects, the gravimetric map was regenerated using only the BRGM and ROB databases, after recalibrating the *Dinant Basin* campaign (1962) and removing the outliers mentioned above. Figure 32 below shows the difference between the latter and the new gravity anomaly map presented in this paper.

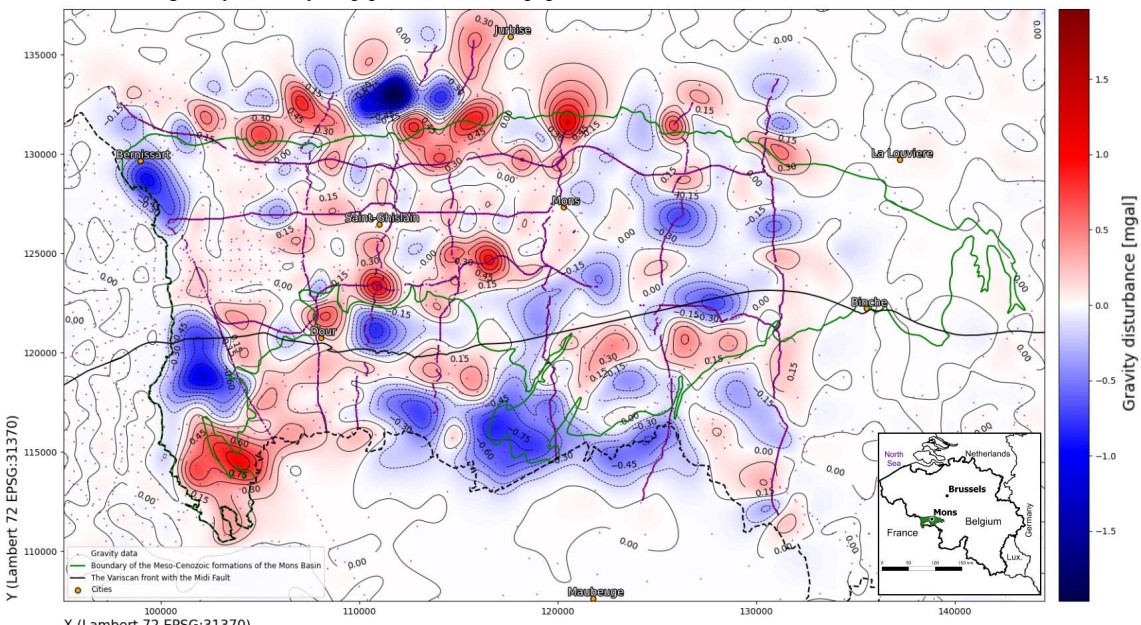

**Figure 32: Difference between the gravity map integrating only the BRGM and ROB databases and the new regional gravity anomaly map (this work).**

This map shows significant differences, ranging from -1.5 to 1.5 mgal. These differences are fairly evenly distributed over the area covered by the MoreGeo and Battaille 1967 campaigns, although the dynamics are somewhat different depending on the location. In fact, the differences are greater at the southern and western margins of the basin than in the centre of the map. This observation is due to the integration of the Battaille 1967 campaign. In fact, this campaign covers large "white" areas close to the French border, which are very poorly covered by the gravimetric campaigns integrated in the BRGM and
ROB databases. On the other hand, the centre of the map shows higher frequency dynamics, mainly along the MoreGeo lines. This observation is related to the hypothesis made above about the accuracy of the Mons (1960) campaign (**Leclerc, 1960**), the only campaign integrated in the ROB database that once covered the centre of the basin.

The impact of these three areas of improvement on the quality and accuracy of gravity mapping in the region is therefore considerable. It is clear that the homogenisation of the spatial distribution of gravity measurements over the whole area, the
acquisition of new high-precision gravity campaigns and the application of topographic corrections have improved this map and made it much more suitable for regional studies. Its refinement is therefore an important step forward in the detection of deep karstic reservoirs. This work will facilitate the use of gravimetric anomalies in this sense, through the production of future gradient maps or their integration into geological modelling processes.

Secondly, it is also interesting to assess the impact of the high spatial sampling rate used in the 2019-2022 MoreGeo





campaign. Indeed, the sampling step set at 50 m between each gravimetric station in this campaign may be too large given the resulting number of data collected across the area. Different anomaly maps were therefore produced, each removing an increasing percentage of randomly selected MoreGeo stations. This process was repeated 50 times in order to extract an average trend in the effect of MoreGeo station removal. Figure 33 shows the results of this compilation, plotting for each percentage of MoreGeo stations removed the average of the mean and maximum differences observed between the latter

and the original map.

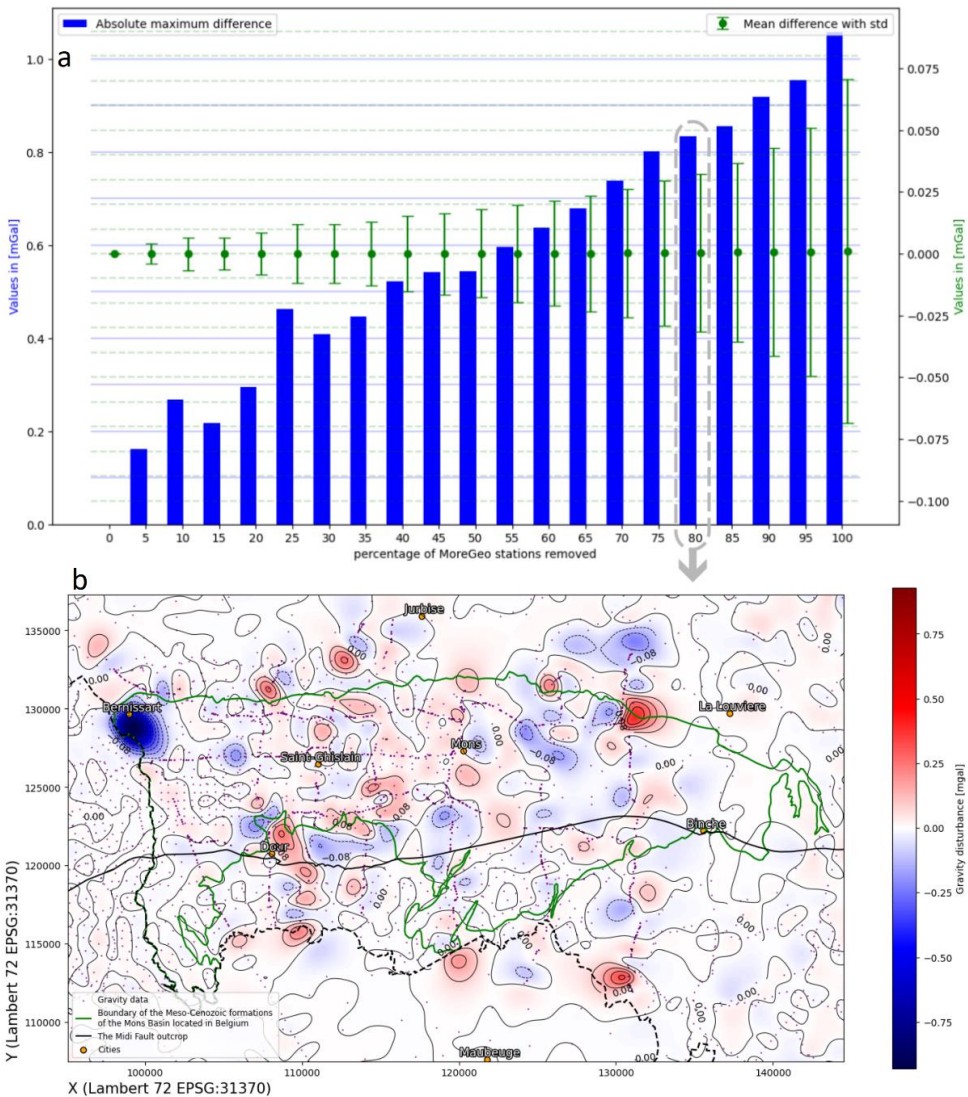

**Figure 33: Impact of MoreGeo station deletion on gravity anomaly mapping results: Comparison between the detailed map and a map generated with a percentage of MoreGeo stations deleted (a). Difference between the new gravity map and the gravity map recalculated with 80% of the MoreGeo stations removed (b).**

As the Fig. 33.a shows, the maximum difference observed between this simulation and the original map (represented by the

blue bar graph and the value scale on the left) increases rapidly in a linear fashion as MoreGeo's sampling density decreases. In fact, deleting 5% of the stations already produces a maximum difference of 0.15 mgal, which exceeds the accuracy initially desired. However, looking at the mean standard deviation of the difference between the simulations and the original map (shown in green with the value scale on the right) gives a different view. This time the evolution is almost exponential and deleting 20% of the MoreGeo stations results in a maximum acceptable deviation of ±0.01 mgal. The map in the




Fig.33.b shows the differences between the original cartography and a cartography produced using only 20% of the MoreGeo stations.

On this map, differences in the order of a few hundred µgal can be seen, once again demonstrating the impact of the dense sampling of this campaign. In particular, the largest difference resulting from the removal of the MoreGeo stations can be seen to the south of Bernissart, in an area that was free of any gravimetric acquisition until 2019. This negative difference
can be explained by the proximity of the Harchies marshes or by the relief of the Palaeozoic basement in this area. Indeed, its summit forms a well-defined depression at this point (**Cornet and Stevens, 1921-1923; Stevens and Marlière, 1944**). On the other hand, it lies to the right of a synform structure with a north-south axis identified in the Palaeozoic basement, at the level of the Upper Carboniferous mined coal (**Dupont 2021**). This structure includes the Bernissart "crans" zone and is therefore possibly related to the dissolution of evaporites in the underlying viscous formations. Nevertheless, it should be
borne in mind that this is still a poorly sampled area (only 1 station). The size of the area affected by this effect is probably not sufficiently constrained and the interpolation has probably extended this local effect.

Finally, it is useful to conclude this section by proposing an interpretation of the new gravity anomaly map by comparing it with the regional geology. Figure 34 shows the isovalue curves of the new gravity anomaly map together with the simplified geological map of the region and the chronostratigraphic column. Figure 34 focuses on the Mons Basin area, where the
isovalue curves of the gravity anomaly are plotted against the thickness map of the Meso-Cenozoic terrains of the Mons Basin, obtained from the Paleozoic roof map of (**Dupont, 2021**).

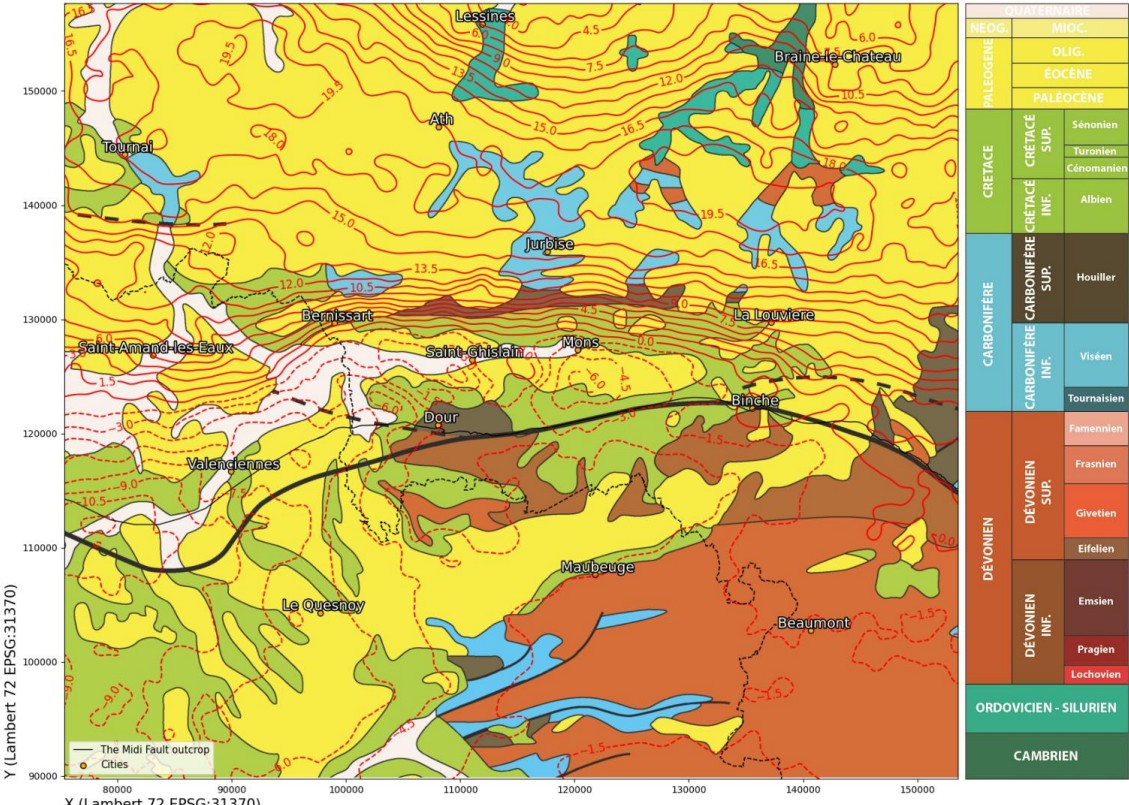

**Figure 34: Iso-value curves of the gravity anomaly with the 1:1,000,000 geological map of France (Chantraine et al., 1996), modified.**

Firstly, a major north-south gradient is immediately apparent, especially in the centre of the map. This major gradient in the
region extends from the west between Nieuport and Dunkirk to the east of Belgium. It is classically interpreted as the northward rise of very dense Palaeozoic pelitic sediments from the Ordovician-Silurian sedimentary cycle, which were



folded during the Caledonian orogeny to form the Brabant Massif **(Everearts and Devos, 2012; Mansy et al., 1999)**. This gravity gradient is currently the most reliable index for locating the Brabant unconformity at depth due to the lack of seismic reflectors. However, in the study area, to the north of the Mons Basin, the gradient is particularly steep, probably

due to the combined effects of the steep uplift of these dense pre-Silurian terranes due to a tectonic wedging effect during the Variscan thrust **(Mansy et al., 1999)** and the thick Meso-Cenozoic sediments of the Mons Basin **(Everaerts and Hennebert, 1998)**. According to **(Everaerts and Hennebert, 1998)** this gradient is also related to the presence of the hypothetical Bordière fault. This large dextral strike-slip fault outcrops to the east of Belgium and is still thought to underlie the Mons region. Its vertical displacement would accentuate the subsidence of the Mons basin **(Legrand, 1968)**.

To the north of this dip, near the towns of Braîne-le-Château and Lessines, the map shows the beginning of the negative Brabant anomaly, which lies directly beneath the Cambrian core of the Brabant Massif. This anomaly, recognised on the Belgian gravity maps, has been interpreted in **(Hennebert, 1993; Mansy et al., 1999)** as the signature of a succession of deep, arc-shaped granitic batholiths, elongated in the direction of the massif and forming a rigid crustal basement. This characteristic negative anomaly was recently used in **(Herbosch and Debacker, 2018)** to map the Brabant Massif under

Meso-Cenozoic cover, in particular by highlighting the Asquempont detachment system, a large extensional fault zone formed shortly after the end of the Ordovician **(Debacker, 2001; Debacker et al., 2004)**.

French Hainaut, which includes the Avesnois region to the south of the map, appears more like a broad gravity trough with a fairly shallow east-west gradient. This area, south of the emergence of the Midi Fault, lies on the edge of the Paris Basin overlying the Palaeozoic terrains of the Ardennes Allochthon **(Belanger et al., 2012)**. This trend is therefore mainly

explained by the thickening of the Meso-Cenozoic sedimentary stack of the Paris Basin and the deepening of the Palaeozoic top towards the west **(Minguely et al., 2006; Mansy et al., 1999)**. In fact, the sinclinal and anticlinal folding structures of the Paleozoic basement of the Ardennes Allochthon, involving Carboniferous and Devonian terrains, do not a priori present sufficiently high density contrasts in this region.

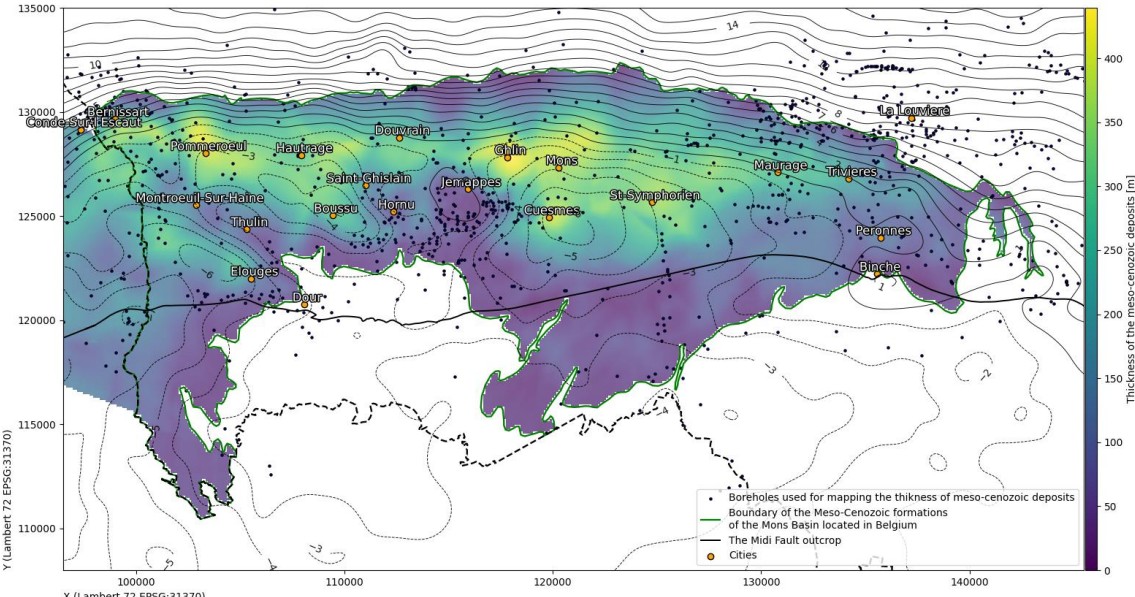

**Figure 35: Iso-value curves of the gravity anomaly with the thickness map of the Meso-Cenozoic formations of the Mons Basin,**
**obtained from the Palaeo-Mesozoic unconformity well data published in (Dupont, 2021).**

A series of negative and positive anomalies are prominent in the centre of the map. In **(Dupont, 2021; Everaerts and Hennebert, 1998; Battaille, 1967)**, these anomalies have already been attributed to variations in the thickness of the Meso-Cenozoic of the Mons Basin by correlating the gravity isovalues with the isohypses at the top of the basement. This new gravity map now allows the lateral extent of these anomalies to be specified and compared with the thickness map of the

Meso-Cenozoic formations in the basin. This is shown in Fig. 35. This figure also shows the location of the towns and



villages where the eponymous structures mentioned below are located. The thickness map in this figure has been generated from the cubic spline interpolation surface of the Palaeo-Mesozoic unconformity well data published in **(Dupont, 2021)** and the high resolution DTM constructed in this work. This thickness map does not include geophysical information.

This overlay shows that the positive and negative gravity anomalies are clearly correlated with the thickness of the Meso-Cenozoic sediments and the relief of the Palaeozoic basement. To the west, two gravity depressions form the contours of the *Sillon d'Élouges* on one side and the *Cuves of Pommeroeul, Herbières (Hautrage) and Boussu* on the other **(Everaerts and Hennebert, 1998; Delmer and Van Wichelen, 1980)**. The positive anomaly framed by these 'basins' can also be divided into two entities: to the south-east the *Surélèvement de Montroeul-sur-Haine - Thulin - Boussu-Bois* and to the north-west the *dome d'Hensies*. This structure has long been mapped, for example in the mapping of the relief of the Palaeozoic

basement in the Quievrain area by **(Cornet and Stevens, 1921-1923; Stevens and Marlière, 1944)**. This is oriented along an axis N130°E, which belongs to the family of main directions N125-145°E of this Meso-Cenozoic unconformity highlighted in **(Licour, 2012; Dupont, 2021)**.

In the centre of the basin, the *Domes d'Hornu* and *Jemappes* form a distinct positive anomaly around the 0 mgal isovalue. The Douvrain basin to the north of these domes is not particularly visible, but it is very likely that the extension of the -1

mgal isovalue in this area of the map is related to the coupled effects of this basin and the proximity of the Palaeozoic outcrop to the north. To the east of this zone, the *Cuves de Cuesmes, Mons and Saint-Symphorien* clearly cause the largest negative anomaly on this map **(Dupont, 2021; Everaerts and Hennebert, 1998)**. At the eastern edge of the basin, more precisely to the north and north-west of Binche, the extension of the 0 and -1 mgal isovalues also appears to be limited by the local overthrusts induced by the *Cuves de Péronnes, Trivières* and *Maurage*.

All these observations and interpretations made on the new map of the Bouguer anomaly are therefore very consistent with the geology, and particularly the shallow geology, of the Mons and Paris sedimentary basins. They also allow the main structures of the Mons Basin, such as the 'Domes' and 'Cuves' and the Brabant Massif, to be located and mapped under cover. These various reasons once again confirm the coherence and accuracy of the methods used in the various stages of this work, such as the gravimetric corrections, the database integration stages and the grid interpolation. This map, and

particularly the dataset used for its production, is therefore an important key for future geological modelling projects, especially that of the Carboniferous limestones of the Brabant Parautochthon, for which the effect of the Meso-Cenozoic terrain must first be removed.

**Author contributions**

QC, ND and OK planned the campaign. QC performed the acquisition campaign. QC and OK processed the data and

developed the gravimetric processing workflow. QC and ND performed the multi-scale fusion of new acquisitions and legacy data. QC performed the regional gravity anomaly mapping and conducted the analysis of the results. QC prepared the manuscript and the figures. OK and ND reviewed and edited the manuscript.

**Competing interests**

The authors declare that they have no conflict of interest

**Acknowledgements**

The authors would like to thank the European Regional Development Fund for its financial support in the framework of the MORE-GEO project.
This work is part of a PhD thesis funded by the Research Institute for Energy and the Research Institute for the Science and Management of Risks of the University of Mons (Faculty of Engineering, Geology and Applied Geology Department).

The authors would also like to thank the researchers Franck Martin, Louis Christiaens and Ivan Nanfo Djoufack for their active participation in the MoreGeo 2019-2022 gravity acquisition campaign.
DeepL Write has been used to improve the clarity and linguistic quality of this document. No content has been generated or modified beyond stylistic adjustments.

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
