# Peer review of "Improving the Gravity Anomaly Map of French-Belgian Hainaut using Multi-Scale Fusion of New Gravity Acquisitions and Legacy Data in an Adaptive and Open-Source Gravimetric Processing Workflow."

_EGUsphere, 2025_

## Referee Comment (RC1)

**Review for manuscript egusphere-2025-497 "Improving the Gravity Anomaly Map of French-Belgian Hainaut using Multi-Scale Fusion of New Gravity Acquisitions and Legacy Data in an Adaptive and Open-Source Gravimetric Processing Workflow"**

This paper mainly presents the acquisition and processing of a new gravity campaign MoreGeo, and how it is combined with existing gravity data in the region to obtain an improved gravity anomaly map. A short discussion was provided in the end regarding how this gravity anomaly map can be used to indicate geological structures. The topic is within the scope of the journal. However, at the current version, the manuscript reads to me more like a technical report documenting the data processing steps. For example, the authors used large content explaining the different types of correction applied (latitude, Bouguer, free-air, terrain correction and weighted least squares), which are already very well-established concept in literature. I would strongly recommend the authors to focus on their own innovative work. What is even more problematic is that these corrections are repeated here but not explained in a clear way, and there are some major mistakes. The figures are commonly difficult to read and presented without providing sufficient explanation and discussion. A large amount of reference are missing, i.e., equations or concepts are given without any references. Please find below a list of detailed comments:

1. P1L30 and throughout the whole manuscript: please correct the citation form.
2. P2Figure1: a) the text within the submap is totally invisible; b) for the same region, colors (gravity anomaly values) in the big map and the submap are different. This needs further explanation.
3. Throughout the paper, the author uses density value (e.g., 2.67 in Figure 1 caption) without unit. But sometimes the density value is used as 2.67 and sometimes as 2670. Please add the unit and keep consistency.
4. P2L37: "The gravity data used to ...available in public databases". Please provide references and where to access the data
5. P3Figure2: Again this figure is really difficult to read, especially the text in Fig2a, the boreholes in Fig2c are invisible, and the caption of Fig2d wrote seismic line L1903. What is L1903 and where is it in Fig2d? This is not explained
6. P3L76: "gravity anomaly that should be observed along these lines based on a density model and a geological model..." why gravity anomaly needs geological model?
7. P4L85: "improving the mapping of the anomaly in a sub-regional sense has many other advantages. For example, it would help to better constrain any future modelling of the geothermal reservoir outside the seismic lines." Please provide some explanation why gravity anomaly mapping can constrain geothermal modeling, it is not clear
8. P4L113: what is "WALCORS GNSS network", explanation or reference needed
9. P4L116: isn't "location of the stations" and "their distribution" the same thing?
10. P4L120 and throughout the paper: the term EPSG:XXXX is used a lot but

without explanation or references.

11. In P4L127: what is acquisition parameters, here it is written "As far as the acquisition parameters are chosen, the most important..." then in P5L133, it is written "Other acquisition parameters..."

12. P5L145-155, this section is difficult to understand.

13. Figure 3: what is the X and Y axis? unit?

14. P6L170 "The table below provides a summary by profile number." I did not see any table

15. P6L175 "three areas along the seismic lines could not be covered by the gravimetric survey." where are the three areas

16. P7L185 "These were implemented using notebooks written in Python 3. A Python module containing all the functions developed and used in these processes was also created." Are the notebooks written by the authors or from others? If it is the latter case, references are needed

17. P7L192 "The model used in this paper is from (Longman, 1959)." Why to use such an old tide model, there are more recent ones and is this old model still suitable?

18. P7L196 "Figure 4 shows the range of this correction, which is between -0.2 and 0.1 mgal..." However, in Fig4, there are clearly values larger than 0.1 mgal

19. P7L199 "This panel also illustrates the 7-day period required to produce this profile." How can this information be seen in Fig 4a

20. P7L201 and throughout the paper: what is et?

21. Equation 1-9: are these equations developed by the authors or from others, references are required in the latte case

22. P10L185 "Although it's much better than the previous one", Why Fig 8b result is now better, please add some explanation. Also why Figure 9 is also better than Fig8b

23. Equation 11-12, 21-23, 29 reference missing

24. P12L323 where does "These 111 loops" coming from

25. P13Fig10b, where do these numbers coming from? Also there are 80 equations in Fig10, why in L346 you wrote "these 84 equations", there are no explanations

26. P13L344: Reference to "graph traversal algorithm" missing

27. P14Fig11, a) the authors claimed that this figure is gravity measurement and used the symbol g which is for absolute gravity. The values of gravity on Earth surface cannot be around 4300 mGal. Please check. b) it is not a good idea to change colors for representing base 1/2/3 between figures, this could be confusing. c) there is no purple color in fig 11b

28. P15L397 What is "outdoor correction"

29. P17L460, if equations are provided, then they should be explained, what are all the symbols in Fig14? They are not explained

30. P20 The X and Y axis of Fig16 and 15 are not consistent

31. P22 Figure19 not properly explained, what are the cyan color dots

32. P24L628, where did the " ge, k and e2" come from? It was not in equation26

33. P25 Equation 27 is wrong, please check the reference. Also the statement following eq27 cannot be true, the effect of atmospheric correction would not be in the order of ugal according to the h

34. P25 Equation28: form g to gravity anomaly, the normal gravity should be removed

35. P26L677 "which is 12 cm, corresponding to a maximum error of about ±37 μgal" how is the 37 uGal obtained?

36. P28Figure26, the authors wrote that this is the raw collection of data for the three surveys. However, there is a mix of usage between absolute gravity value g (value ranging around 981100) and likely gravity anomaly/disturbance (value -15 to 2).

37. P29Figure27, also Figure28: These values in the figure cannot be gravity disturbance, they should be absolute gravity values

38. P32Figure30: it is difficult to read any information from the figure, please make it more clear

39. P32L839, where does the "R^2" coming from?

40. P33 L865 writes that Fig31 presents the Bouguer anomaly but in the figure it is written gravity disturbance. Please check and correct the usage between gravity, gravity anomaly and gravity disturbance throughout the paper.

---

## Referee Comment (RC2)

**Paper review for EGUsphere**

**Title:** Improving the Gravity Anomaly Map of French-Belgian Hainaut using Multi-Scale Fusion of New Gravity Acquisition and Legacy Data in an Adaptive and Open-Source Gravimetric Processing Workflow.

**Authors:** Q. Campeol, Dupont N and Kaufmann O.

**Result of the review:** worthy of publishing, but **substantial revision** is needed

**General comments:** As a lecturer of Physical Geodesy for many years, active participant in numerous international gravimetric surveys, manager of large regional gravity database that includes data from different countries and as active user of gravity data for both geodetic and geophysical purposes, I'm truly puzzled by this contribution. Many modeling problems mentioned in the paper look familiar, however the way that these problems are addressed is often non-standard. In other words, the data processing elements (the corrections) that you mention in the paper are completely understandable-, important- and worth addressing, but the way you address them does not always follow what is "the common practice" among the practitioners.

To my opinion the paper suffers, technically, from "too much irrelevant information" that blur the content. For example, in your writing it is not necessary to refer to pre-computer age graphical methods of terrain correction and to Hammer zones (as if it was an option). You should focus on what you did in practice to correct the gravitational effect of the near and far terrain. To my opinion, and because of this, the main message of what you did with MoreGeo data drowns in all kinds of irrelevant detail. Also, the prism formula for the gravitational potential of Nagy et al., 2000, 2002 (which you didn't use) is also irrelevant. Terrain corrections, Bouguer anomalies (both simple and refined) are part of standard corrections in Physical Geodesy and in gravity surveys and must not be explained "from scratch".

My experience from many years of international gravity projects is that different countries have slightly different approaches to how to design and conduct a gravity survey. In this context I'm also aware of different approaches across different disciplines of geosciences - between geodesy and geophysics/geology. When we include gravity data to a gravity database, we sometimes face a problem that the data collected for the purpose of geology/geophysics are not always properly tied to a well-defined absolute system. In your paper you claim a connection to the world system (GRS80 ellipsoid and WGS84 datum through your RTK positioning). However, GRS80 is more than the geometrical aspect. As geodesist there are two things in your paper that bother me concerning your explanations of the MoreGeo project processing: (1) Where are the absolute gravity references/ties of your measurements? (2) Why is the normal gravity model (GRS80) not mentioned explicitly? This is important and probably hidden in your Eq. (26) where you quote (Moritz 1980). A standard definition of free-air gravity anomalies is by using a normal gravity model (currently GRS80). In your contribution it is (probably) what you call, the latitude correction. By definition, a gravity anomaly means a difference between the true gravity value (measured by the relative gravimeter; g_measured in your Eq. 28) and a normal gravity value $\gamma$ of the normal gravity model. Thus, a true gravity field is treated as anomaly with respect to the (normal) model field. This aspect is not explicit in your writing.

Also, I'm completely amazed by your way of treating the collected gravity data. You seem to correct directly the gravity readings until you obtain the anomalies. On the positive side, the survey results that you obtain and show on Figure 12 (green dots) look fine to me. The problem with this figure is the numbers. Your y-axis on Fig 9, Fig. 11 and Fig. 12 is not (as you claim) "g

[mgal]" but the reduced gravity readings (?). "g" is reserved for absolute gravity and 6-digit numbers (when in mGals) starting with "9". I do not know any place on Earth where the Bouguer gravity anomaly is 4300 mGals as it is on most of your figures!

Another thing that truly amazes me is the idea of modeling the relative gravimeter drift over large time span to ensure (as I understand it) that the drift corrected gravity readings yield the same value at points that the survey measured more than once (possibly at entirely different times). The drift model refers to the specific time (the start time of the first period) and the corrected readings express what the gravimeter would measure at all points of the profile if the gravimeter drift was zero. However, at the level of accuracy that you aim for (sub 10 microgals) there can be environmental changes that can occur around the stations between the two times so that the local reading is affected (e.g. hydrology). For modelling the long term gravimeter drift the paper that you are inspired by a paper on drift of a superconducting gravimeter which is a stationary instrument measuring over large time span and affected by the same gravitational pull of the Earth for this location. For such instrument it makes sense to consider a drift model over long time, but not necessarily for a relative gravimeter like CG-6.

You have completely different setup for the relative gravimeters and a microgravity survey that you conduct. If I was to plan such a microgravity survey, I would have included reference points with absolute gravity values and (if possible) visit on the way other points with known and well-established absolute gravity values. I realize that you measure 40 points a day which makes it logistically difficult. But this is about independent validation of your drift model. In this setup, the drift would be determined daily and would for a given day of measurements approximately be linear. After all, there is a drift self-calibration option a CG6. One could almost do daily such drift calibration test overnight. Consequently, you can determine CG6 drift daily and compare it to the one obtained from the absolute stations. In fact, you do not even need an absolute station in the field to get a drift estimate. You just need a base point that you start from and end at the same day. On the positive side, you seem to get something reasonable with the long time span drift function.

A different (but serious) technical issue with the paper is the notation and your very loose use of different concepts. Equations (21), (22) and (23) make no sense. Please, look them up in (Heiskanen and Moritz, 1967) that you quote. Bouguer gravity anomaly, free-air gravity anomaly, gravity disturbances have precise meaning. They need to be defined first. Use consistent symbols. In Eq. (25) you write $rho$ for density and $\rho$ in prism formulas just below. I'm also in doubt about your loose terminology regarding heights. Ellipsoidal heights are usually denoted $h$ while normal- and orthometric heights are denoted $H^*$ and $H$ respectively. It seems you only use ellipsoidal heights, but I don't know. You also get the free-air reduction wrong. Yes, I do realize that there is a minor latitude effect as stated in your Eq. (24). Your normal gravity $\gamma$ as a function of ellipsoidal height is for all terrestrial points approximated as Taylor series in powers of $h$ truncated at degree 2. When you differentiate wrt. $h$ to get $\dfrac{\partial \gamma}{\partial h}$ you should get a 1 order polynomial. Free-air reduction $F$ uses $-\dfrac{\partial g}{\partial H}H \approx -\dfrac{\partial \gamma}{\partial h}H$ so that $F = 0.3086 \cdot H$ (mGal) when $H$ is in meters. Unfortunately, in (Heiskanen and Moritz, 1967) there is a notational inconsistency in Eq. (3-17) where they write: $F = 0.3086 \cdot h$, but what they mean by h in this equation is the orthometric height (height over geoid). This inconsistency is corrected in (Hofmann-Wellenhof

and Moritz, 2006), eq. (3-26). Bottom line, it is $F \equiv -\dfrac{\partial \gamma}{\partial h} H$ and not $F \equiv -\dfrac{\partial \gamma}{\partial h} h$ which corresponds to your eq.(24), The base level of the Bouguer plate or Bouguer spherical shell is geoid/quasi-geoid and not the ellipsoid! Referring to your Eq. (24) it is only the first part of the equation that is used as a standard and for $H$ as a station height not $h$ . Other examples of strange notation in the paper are Eq. (7), (8) and (9) denoting (I guess) confidence intervals(?) of student distribution. Where is LHS (left-hand-side) of the equation? What does it mean?

**What to do?**

Let me be clear, I encourage you to correct the paper. I find the results and this kind of cross border reconciliation of gravity data important and relevant. Unfortunately, if I do not understand in full what you did with GeoMore data, the question of merging these results with legacy data is at this stage irrelevant. As it is now, I'm not entirely sure if you and I even agree on what is free-air gravity anomaly, Bouguer gravity anomaly (simple, refined), terrain correction, topographic correction … etc. Your paper in its current state is, to put it mildly, not helpful. So, before you and I can engage in any meaningful discussion about the content, these things must be corrected. Reduce the paper considerably, use appendices for details, use strict and standard definitions, use systematic and uniform standard mathematical notation, be explicit about the heights that you use at each step (ellipsoidal, orthometric, normal), relate the introduced concepts to global models (GRS80 normal gravity model). If you still insist on formulating your processing steps as corrections relate them to the above standard definition of concepts. Use the absolute gravity ties explicitly. Be very careful about your figures, the numbers that you show and how you label your y-axis.

Concerning the last part of the paper, the consistency with the legacy gravity data, there are important topics to discuss (e.g. reconciling different conventions of reducing the data sets, truncated station location coordinates, map projection issues, different conventions regarding topographic mass density assumptions … etc ). But we cannot engage in this kind of discussion before the paper is corrected.

Good luck with it!

---

## Author Comment (AC1)

**Response for review of the manuscript egusphere-2025-497 "Improving the Gravity Anomaly Map of French-Belgian Hainaut using Multi-Scale Fusion of New Gravity Acquisitions and Legacy Data in an Adaptive and Open-Source Gravimetric Processing Workflow"**

XXX : Anonymous Referee #1
XXX : Authors

This paper mainly presents the acquisition and processing of a new gravity campaign MoreGeo, and how it is combined with existing gravity data in the region to obtain an improved gravity anomaly map. A short discussion was provided in the end regarding how this gravity anomaly map can be used to indicate geological structures. The topic is within the scope of the journal. However, at the current version, the manuscript reads to me more like a technical report documenting the data processing steps. For example, the authors used large content explaining the different types of correction applied (latitude, Bouguer, free-air, terrain correction and weighted least squares), which are already very well-established concept in literature. I would strongly recommend the authors to focus on their own innovative work. What is even more problematic is that these corrections are repeated here but not explained in a clear way, and there are some major mistakes. The figures are commonly difficult to read and presented without providing sufficient explanation and discussion. A large amount of reference are missing, i.e., equations or concepts are given without any references. Please find below a list of detailed comments:

Authors would like to thank you for your detailed comments and for the time you have spent commenting this submission, which further motivates its revision. Authors will reorganise the text to reduce the amount of information and improve the way it is presented.
With regard to the section devoted to processing and, in particular, gravimetric corrections, authors feel that it is appropriate to include it in the article, given its involvement in the gravimetric processing that authors have carried out. However, authors admit that its current form requires major modifications and that certain parts can be moved to the appendix. Necessary references will be added and concepts that are more than established in the literature will be synthesised or simply quoted, in accordance with your comments. With regard to the figures, authors will pay particular attention to their legibility, their number and their discussion in the text, also in accordance with your comments.

1. P1L30 and throughout the whole manuscript: please correct the citation form
- The citation form will be corrected throughout the manuscript.

2. P2Figure1: a) the text within the submap is totally invisible; b) for the same region, colors (gravity anomaly values) in the big map and the submap are different. This needs further explanation.
- The submap legend will be enlarged to make it visible. The big map and the submap come from different authors, which is why the colorbars don't quite match. This will be explained in the text.

3. Throughout the paper, the author uses density value (e.g., 2.67 in Figure 1 caption) without unit. But sometimes the density value is used as 2.67 and sometimes as 2670. Please add the unit and keep consistency.
- The density value of 2.67 will be used throughout the text.

4. P2L37: "The gravity data used to ...available in public databases". Please provide references and where to access the data
- The DOI for the reference (Verbeurgt et al., 2019) (https://doi.org/10.24414/j0dx-9n36) and the WMS link for the reference (BRGM, 2023) (http://geoservices.brgm.fr/geologie) will be added to the bibliography which permit to access the data.

5. P3Figure2: Again this figure is really difficult to read, especially the text in Fig2a, the boreholes in Fig2c are invisible, and the caption of Fig2d wrote seismic line L1903. What is L1903 and where is it in Fig2d? This is not explained
- The legibility of this figure will be improved, either by increasing the font size or by placing a larger version in landscape format in the appendix. The location in the Fig2d section of the seismic section

L190. will be better explained on the map in Fig2c.

6. P3L76: "gravity anomaly that should be observed along these lines based on a density model and a geological model..." why gravity anomaly needs geological model?

• The idea behind this former work was to demonstrate the inconsistency between the former gravity anomaly map from (Everaerts and De Vos, 2012) and the interpretation of the 2012 seismic sections (Dupont, 2021). This highlighted the need to resample the gravity anomaly more finely in the area and probably to redesign the regional deep geological model (P3L80).

7. P4L85: "improving the mapping of the anomaly in a sub-regional sense has many other advantages. For example, it would help to better constrain any future modelling of the geothermal reservoir outside the seismic lines." Please provide some explanation why gravity anomaly mapping can constrain geothermal modeling, it is not clear

• The link between gravity anomaly mapping and geological modelling for geothermal energy is introduced in P2L55 but I agree a reminder would be appropriate here, specifying that this would help to constrain the location of karstified levels and those of massive anhydrites within the deep reservoir.

8. P4L113: what is "WALCORS GNSS network", explanation or reference needed

• Authors will add reference about WALCORS GNSS network.

9. P4L116: isn't "location of the stations" and "their distribution" the same thing?

• This line will be modified as "...defining the location of the profiles on the surface of the study area and the spatial distribution of the stations within them.".

10. P4L120 and throughout the paper: the term EPSG:XXXX is used a lot but without explanation or references.

• The definition of the acronym EPSG will be added at the first mention in the text.

11. In P4L127: what is acquisition parameters, here it is written "As far as the acquisition parameters are chosen, the most important..." then in P5L133, it is written "Other acquisition parameters…"

• The acquisition parameters are all the parameters mentionned in this part, between P4L127 and P4L137.

12. P5L145-155, this section is difficult to understand.

• A reorganisation and refomulation of this part will be carried out.

13. Figure 3: what is the X and Y axis? Unit?

• The unit is in meter due to the CRS Lambert 72 mentionned in both axes. This is also the reason why there is no graphic scale on the maps where this indication appears on the X and Y axes. To prevent any confusion, an explanation will be added in the text when this map style appears.

14. P6L170 "The table below provides a summary by profile number." I did not see any table

• Indeed, there is a problem between two versions of the document. The final version should no longer contain a summary table. This sentence will be deleted.

15. P6L175 "three areas along the seismic lines could not be covered by the gravimetric survey." where are the three areas

• Few details on the location of these zones will be added.

16. P7L185 "These were implemented using notebooks written in Python 3. A Python module containing all the functions developed and used in these processes was also created." Are the notebooks written by the authors or from others? If it is the latter case, references are needed

• All notebooks were written by authors, this comment will be added.

17. P7L192 "The model used in this paper is from (Longman, 1959)." Why to use such an old tide model, there are more recent ones and is this old model still suitable?

• Although there are more recent models ( Hartmann & Wenzel (1995) or Tamura (1987) ), it is all a question of the accuracy of the correction. Some recent paper (Trabanco, Jorge and Rogério Rodrigues Amarant (2016). Calculation of the tide correction used in gravimetry. Revista Brasileira de Geofísica.) evaluate that for the longman tide model, "some computer programs adopt equivalent values and do not yield significant results". Compared to the initial target of 100 µGal, the longman tide model seems correct but this still needs to be investigated and commented on.

18. P7L196 "Figure 4 shows the range of this correction, which is between -0.2 and 0.1 mgal..." However, in Fig4, there are clearly values larger than 0.1 mgal

- That's true. This is due to confusion over the references used between the text and the figure. The text refers to the absolute tide correction range over all the survey, whereas the figure illustrates this range minus the value at the base station used for the corrections, which shifts the range window by +30µgal. This confusion will be corrected.

19. P7L199 "This panel also illustrates the 7-day period required to produce this profile." How can this information be seen in Fig 4a

- 7 is a error, 5 days is the actual time required to produce this profil and this can be seen by the 5 sets of green lines only on FIG4a. Nevertheless, this information is not considered necessary and will be removed.

20. P7L201 and throughout the paper: what is et?

- Indeed, there is a typo in the text, 'et' = 'and' in French, it will be corrected.

21. Equation 1-9: are these equations developed by the authors or from others, references are required in the latte case

- These equations were written by authors.

22. P10L185 "Although it's much better than the previous one", Why Fig 8b result is now better, please add some explanation. Also why Figure 9 is also better than Fig8b

- Some explanation about this comment will be added.

23. Equation 11-12, 21-23, 29 reference missing

- References will be added.

24. P12L323 where does "These 111 loops" coming from

- These loops are pairs of acquisitions made at the same station in different periods and were used to reposition the 5 periods in relation to a reference period. The number of loops is not important. This section will be reworded.

25. P13Fig10b, where do these numbers coming from? Also there are 80 equations in Fig10, why in L346 you wrote "these 84 equations", there are no explanations

- These numbers come from drift relationships between periods for the MoreGeo survey (P13L340 and shown in FIG10a). This is probably a typo, 80 is the correct number of relationships kept after filtering.

26. P13L344: Reference to "graph traversal algorithm" missing

- Reference will be added.

27. P14Fig11, a) the authors claimed that this figure is gravity measurement and used the symbol g which is for absolute gravity. The values of gravity on Earth surface cannot be around 4300 mGal. Please check. b) it is not a good idea to change colors for representing base 1/2/3 between figures, this could be confusing. c) there is no purple color in fig 11b

- All these comments will be taken into account by authors, the figure and its legend will be corrected.

28. P15L397 What is "outdoor correction"

- This typo will be replaced by free-air correction.

29. P17L460, if equations are provided, then they should be explained, what are all the symbols in Fig14? They are not explained

- As proposed by referee no. 2, this point should be deleted from the text because no use is made of it afterwards.

30. P20 The X and Y axis of Fig16 and 15 are not consistent

- The X and Y axes in FIG16 and FIG15b are different because the grids are georeferenced in different CRS. In particular, the Pseudo-Mercator projection has been used for the regional terrain correction elevation grid, as the Lambert 1972 projection, which is generally used in Belgium, would lead to excessive distortions. Nevertheless, the authors will add the unit [m] and comments on this subject.

31. P22 Figure19 not properly explained, what are the cyan color dots

- The authors will take care to improve the comprehensibility of this figure.

32. P24L628, where did the " ge, k and e2" come from? It was not in equation26

- These notations come from an earlier version of the document and have been replaced by A, B and C for ease of reading. This typo will be corrected.

33. P25 Equation 27 is wrong, please check the reference. Also the statement following eq27 cannot be true, the effect of atmospheric correction would not be in the order of ugal according to the h

- The authors will correct this error. 10e-5 * h and 10e-9 * h² have been replaced by 10-5h and 10-9h² respectively when writing the Word equations.

34. P25 Equation28: form g to gravity anomaly, the normal gravity should be removed

- Authors will replaced g by Instrumental measurement in the P25 Equation28.

35. P26L677 "which is 12 cm, corresponding to a maximum error of about ±37 µgal" how is the 37 uGal obtained?

- This value, which was also announced during the presentation of the 1x1 m² Walloon DTM for 2013-2014 (P20L522), comes from the owner of the data, the Service Public de Wallonie (SPW) (https://geoportail.wallonie.be).

36. P28Figure26, the authors wrote that this is the raw collection of data for the three surveys. However, there is a mix of usage between absolute gravity value g (value ranging around 981100) and likely gravity anomaly/disturbance (value -15 to 2).

- Indeed, and this is illustrated expressly to show one of the difficulties of having fused the different data sets. However, your comment will be added in the figure label to avoid any confusion.

37. P29Figure27, also Figure28: These values in the figure cannot be gravity disturbance, they should be absolute gravity values

- Absolutely, authors will replace gravity disturbance with absolute gravity in these figures.

38. P32Figure30: it is difficult to read any information from the figure, please make it more clear

- The figure will be enlarged and moved to the appendix to make it visible.

39. P32L839, where does the "R^2" coming from?

- A commentary on the R^2 calculation will be added to the text. It is from the average of the squared differences between the observed anomaly and the anomaly simulated by equivalent sources.

40. P33 L865 writes that Fig31 presents the Bouguer anomaly but in the figure it is written gravity disturbance. Please check and correct the usage between gravity, gravity anomaly and gravity disturbance throughout the paper.

- These notions will be checked and corrected throughout the paper.

---

## Author Comment (AC2)

**See the author's response above**

**Paper review for EGUsphere**

**Title:** Improving the Gravity Anomaly Map of French-Belgian Hainaut using Multi-Scale Fusion of New Gravity Acquisition and Legacy Data in an Adaptive and Open-Source Gravimetric Processing Workflow.

**Authors:** Q. Campeol, Dupont N and Kaufmann O.

**Result of the review:** worthy of publishing, but substantial revision is needed

**General comments:** As a lecturer of Physical Geodesy for many years, active participant in numerous international gravimetric surveys, manager of large regional gravity database that includes data from different countries and as active user of gravity data for both geodetic and geophysical purposes, I'm truly puzzled by this contribution. Many modeling problems mentioned in the paper look familiar, however the way that these problems are addressed is often non-standard. In other words, the data processing elements (the corrections) that you mention in the paper are completely understandable-, important- and worth addressing, but the way you address them does not always follow what is "the common practice" among the practitioners.

To my opinion the paper suffers, technically, from "too much irrelevant information" that blur the content. For example, in your writing it is not necessary to refer to pre-computer age graphical methods of terrain correction and to Hammer zones (as if it was an option). You should focus on what you did in practice to correct the gravitational effect of the near and far terrain. To my opinion, and because of this, the main message of what you did with MoreGeo data drowns in all kinds of irrelevant detail. Also, the prism formula for the gravitational potential of Nagy et al., 2000, 2002 (which you didn't use) is also irrelevant. Terrain corrections, Bouguer anomalies (both simple and refined) are part of standard corrections in Physical Geodesy and in gravity surveys and must not be explained "from scratch".

My experience from many years of international gravity projects is that different countries have slightly different approaches to how to design and conduct a gravity survey. In this context I'm also aware of different approaches across different disciplines of geosciences - between geodesy and geophysics/geology. When we include gravity data to a gravity database, we sometimes face a problem that the data collected for the purpose of geology/geophysics are not always properly tied to a well-defined absolute system. In your paper you claim a connection to the world system (GRS80 ellipsoid and WGS84 datum through your RTK positioning). However, GRS80 is more than the geometrical aspect. As geodesist there are two things in your paper that bother me concerning your explanations of the MoreGeo project processing: (1) Where are the absolute gravity references/ties of your measurements? (2) Why is the normal gravity model (GRS80) not mentioned explicitly? This is important and probably hidden in your Eq. (26) where you quote (Moritz 1980). A standard definition of free-air gravity anomalies is by using a normal gravity model (currently GRS80). In your contribution it is (probably) what you call, the latitude correction. By definition, a gravity anomaly means a difference between the true gravity value (measured by the relative gravimeter; g_measured in your Eq. 28) and a normal gravity value $\gamma$ of the normal gravity model. Thus, a true gravity field is treated as anomaly with respect to the (normal) model field. This aspect is not explicit in your writing.

Also, I'm completely amazed by your way of treating the collected gravity data. You seem to correct directly the gravity readings until you obtain the anomalies. On the positive side, the survey results that you obtain and show on Figure 12 (green dots) look fine to me. The problem with this figure is the numbers. Your y-axis on Fig 9, Fig. 11 and Fig. 12 is not (as you claim) "g

[mgal]" but the reduced gravity readings (?). "g" is reserved for absolute gravity and 6-digit numbers (when in mGals) starting with "9". I do not know any place on Earth where the Bouguer gravity anomaly is 4300 mGals as it is on most of your figures!

Another thing that truly amazes me is the idea of modeling the relative gravimeter drift over large time span to ensure (as I understand it) that the drift corrected gravity readings yield the same value at points that the survey measured more than once (possibly at entirely different times). The drift model refers to the specific time (the start time of the first period) and the corrected readings express what the gravimeter would measure at all points of the profile if the gravimeter drift was zero. However, at the level of accuracy that you aim for (sub 10 microgals) there can be environmental changes that can occur around the stations between the two times so that the local reading is affected (e.g. hydrology). For modelling the long term gravimeter drift the paper that you are inspired by a paper on drift of a superconducting gravimeter which is a stationary instrument measuring over large time span and affected by the same gravitational pull of the Earth for this location. For such instrument it makes sense to consider a drift model over long time, but not necessarily for a relative gravimeter like CG-6.

You have completely different setup for the relative gravimeters and a microgravity survey that you conduct. If I was to plan such a microgravity survey, I would have included reference points with absolute gravity values and (if possible) visit on the way other points with known and well-established absolute gravity values. I realize that you measure 40 points a day which makes it logistically difficult. But this is about independent validation of your drift model. In this setup, the drift would be determined daily and would for a given day of measurements approximately be linear. After all, there is a drift self-calibration option a CG6. One could almost do daily such drift calibration test overnight. Consequently, you can determine CG6 drift daily and compare it to the one obtained from the absolute stations. In fact, you do not even need an absolute station in the field to get a drift estimate. You just need a base point that you start from and end at the same day. On the positive side, you seem to get something reasonable with the long time span drift function.

A different (but serious) technical issue with the paper is the notation and your very loose use of different concepts. Equations (21), (22) and (23) make no sense. Please, look them up in (Heiskanen and Moritz, 1967) that you quote. Bouguer gravity anomaly, free-air gravity anomaly, gravity disturbances have precise meaning. They need to be defined first. Use consistent symbols. In Eq. (25) you write $rho$ for density and $\rho$ in prism formulas just below. I'm also in doubt about your loose terminology regarding heights. Ellipsoidal heights are usually denoted $h$ while normal- and orthometric heights are denoted $H^*$ and $H$ respectively. It seems you only use ellipsoidal heights, but I don't know. You also get the free-air reduction wrong. Yes, I do realize that there is a minor latitude effect as stated in your Eq. (24). Your normal gravity $\gamma$ as a function of ellipsoidal height is for all terrestrial points approximated as Taylor series in powers of $h$ truncated at degree 2. When you differentiate wrt. $h$ to get $\dfrac{\partial \gamma}{\partial h}$ you should get a 1 order polynomial. Free-air reduction $F$ uses $-\dfrac{\partial g}{\partial H}H \approx -\dfrac{\partial \gamma}{\partial h}H$ so that $F = 0.3086 \cdot H$ (mGal) when $H$ is in meters. Unfortunately, in (Heiskanen and Moritz, 1967) there is a notational inconsistency in Eq. (3-17) where they write: $F = 0.3086 \cdot h$, but what they mean by h in this equation is the orthometric height (height over geoid). This inconsistency is corrected in (Hofmann-Wellenhof

and Moritz, 2006), eq. (3-26). Bottom line, it is $F \equiv -\frac{\partial \gamma}{\partial h} H$ and not $F \equiv -\frac{\partial \gamma}{\partial h} h$ which corresponds to your eq.(24), The base level of the Bouguer plate or Bouguer spherical shell is geoid/quasi-geoid and not the ellipsoid! Referring to your Eq. (24) it is only the first part of the equation that is used as a standard and for $H$ as a station height not $h$ . Other examples of strange notation in the paper are Eq. (7), (8) and (9) denoting (I guess) confidence intervals(?) of student distribution. Where is LHS (left-hand-side) of the equation? What does it mean?

**What to do?**

Let me be clear, I encourage you to correct the paper. I find the results and this kind of cross border reconciliation of gravity data important and relevant. Unfortunately, if I do not understand in full what you did with GeoMore data, the question of merging these results with legacy data is at this stage irrelevant. As it is now, I'm not entirely sure if you and I even agree on what is free-air gravity anomaly, Bouguer gravity anomaly (simple, refined), terrain correction, topographic correction … etc. Your paper in its current state is, to put it mildly, not helpful. So, before you and I can engage in any meaningful discussion about the content, these things must be corrected. Reduce the paper considerably, use appendices for details, use strict and standard definitions, use systematic and uniform standard mathematical notation, be explicit about the heights that you use at each step (ellipsoidal, orthometric, normal), relate the introduced concepts to global models (GRS80 normal gravity model). If you still insist on formulating your processing steps as corrections relate them to the above standard definition of concepts. Use the absolute gravity ties explicitly. Be very careful about your figures, the numbers that you show and how you label your y-axis.

Concerning the last part of the paper, the consistency with the legacy gravity data, there are important topics to discuss (e.g. reconciling different conventions of reducing the data sets, truncated station location coordinates, map projection issues, different conventions regarding topographic mass density assumptions … etc ). But we cannot engage in this kind of discussion before the paper is corrected.

Good luck with it!

**Response from the author:**

Firstly, the authors would like to thank you for your detailed comments and for the time you have devoted to them. Your comments and suggestions for improvement will undoubtedly help to enhance the quality of this work. As you wrote, the basic objectives and approach adopted by geophysicists are quite different from those adopted by geodesists. This leads to a different conception of data acquisition and processing, which is reflected in the way the data is presented. However, as the authors wish to integrate MoreGeo data into regional gravimetric databases, your advice is both valuable and necessary. It is therefore advisable to clarify the various concepts used and reorganise the presentation. In this respect, the authors confirm that they are keen to respect 'common practice' provided it does not divert this writing from its original purpose.

Regarding the absolute gravity references/liaisons of the measurements, it is true that it would have been preferable to integrate one during the campaign. However, in the MoreGeo region, the existing absolute gravity values prior to the campaign are relative gravimetry data recalibrated to absolute in the Royal Observatory of Belgium's (ROB) database. These date back to the 1960s. The quality of these measurements is limited by factors such as uncertainty, calibration and, above all, geolocation (one of the authors had to geolocate a large proportion of the data using old maps), as well as regional mining subsidence. As the authors do not own an absolute gravimeter, it was envisaged that they would carry out a calibration using absolute measurements made by an absolute gravimeter owned by the

ROB. However, circumstances did not permit this, so the authors developed the approach presented in this manuscript: positioning stations close to several old absolute gravitational references to recalibrate MoreGeo's measurements a posteriori on absolute gravitational data. P29L743 also states that this calibration was based on the two closest pairs of MoreGeo and ROB stations (5.3 m and 6.8 m apart) and two areas unaffected by mining subsidence close to the Franco-Belgian border. These stations are considered to be the two absolute gravitational references for setting MoreGeo. Nevertheless, the authors acknowledge that a comment on this point should be added to the 'Design of the new surveys' section.

With regard to the normal gravity model, the authors will integrate it into the beginning of the text in order to define the gravity anomaly precisely and explain how it is obtained. The latitude correction Eq. (26) refers to the WGS84 normal gravity model. To avoid confusion, this correction will be replaced by the concept of normal gravity $\gamma$, derived from a normal gravity model. Certain concepts, such as gravity disturbance, gravity anomaly, normal and orthometric heights, and the quasigeoid, also require clarification. The titles of the axes in figures (4), (5), (8), (9), (11), (12), (22), (23), (27), (28), (30), (32) and (33), and equations (7), (8), (9), (21), (22), (23), (24), (25), (26) and (28) will also be revised.

Your comments on the heights used in the various corrections are excellent and have prompted a great deal of thought. The authors would also like to thank you for the reference to (Hofmann-Wellenhof and Moritz, 2006), which will be incorporated into the manuscript. Thanks to this, the authors will be able to clear up the confusion that has crept into the text. The altitudinal reference used to reduce MoreGeo's relative gravity measurements is not the GRS80 ellipsoid, despite what was unfortunately mentioned in P15L404. While the GPS readings of the gravity stations were taken in the WGS84 system based on the GRS80 ellipsoid, the two grids used for the terrain corrections are referenced to the EVRF2000 quasigeoid (P19L508 and P20L537). To ensure consistency, the heights of the MoreGeo stations have been reprojected and referenced to the EVRF2000 quasigeoid, meaning that all corrections now use the normal height $H^*$ relative to this quasigeoid. Therefore, the base level used is the quasigeoid in terrain, free-air and atmospheric corrections, contrary to what was stated in the manuscript and equations. Reduced gravity measurements $g(P0)$, related to the European quasigeoid, are used to calculate the gravity anomaly $\Delta g$ by comparing them with the WGS84 normal gravity $\gamma(Q0)$ of Eq. (26), as shown in equations (2-228) and (8-127) of (Hofmann-Wellenhof and Moritz, 2006). The authors will amend the text accordingly (in particular, regarding P20L537: 'EVRF2000 is a reference frame related to the quasigeoid, not an ellipsoid').

The authors would also like to thank you for your invaluable, practical advice on carrying out a gravimetric acquisition that incorporates daily drift correction. This will be very useful for future acquisitions. Unfortunately, these tips were not applicable to MoreGeo, as it was carried out in two phases (2019 and 2022) by different teams, and the bases used varied depending on the phase and the operators. Therefore, a solution was implemented that integrated all stations visited more than once at different times, as well as a series of global resampling at the end of 2022.

Your concerns about the expected accuracy due to possible environmental changes are entirely legitimate. Nevertheless, the site was chosen to minimise the impact of these changes, particularly the hydrogeological effects. It is free of vegetation, the thickness of the unsaturated zone is less than two metres and the water level is very stable. Seasonal environmental effects were avoided at the other stations by dividing the campaign into five periods. These effects have therefore been disregarded. These explanations will be added to the text.

In summary, the authors would like to thank you again for your detailed comments. They will make every effort to improve the manuscript. They will also endeavour to reduce the size of the document by transferring certain parts to the appendices.